# Spatial structure impacts adaptive therapy by shaping intra-tumoral competition

Maximilian A. R. Strobl [1,2✉], Jill Gallaher [1], Jeffrey West [1], Mark Robertson-Tessi [1], Philip K. Maini [2,3] & Alexander R. A. Anderson [1,3✉]

## Abstract

**Background** Adaptive therapy aims to tackle cancer drug resistance by leveraging resource competition between drug-sensitive and resistant cells. Here, we present a theoretical study of intra-tumoral competition during adaptive therapy, to investigate under which circumstances it will be superior to aggressive treatment.

**Methods** We develop and analyse a simple, 2-D, on-lattice, agent-based tumour model in which cells are classified as fully drug-sensitive or resistant. Subsequently, we compare this model to its corresponding non-spatial ordinary differential equation model, and fit it to longitudinal prostate-specific antigen data from 65 prostate cancer patients undergoing intermittent androgen deprivation therapy following biochemical recurrence.

**Results** Leveraging the individual-based nature of our model, we explicitly demonstrate competitive suppression of resistance during adaptive therapy, and examine how different factors, such as the initial resistance fraction or resistance costs, alter competition. This not only corroborates our theoretical understanding of adaptive therapy, but also reveals that competition of resistant cells with each other may play a more important role in adaptive therapy in solid tumours than was previously thought. To conclude, we present two case studies, which demonstrate the implications of our work for: (i) mathematical modelling of adaptive therapy, and (ii) the intra-tumoral dynamics in prostate cancer patients during intermittent androgen deprivation treatment, a precursor of adaptive therapy.

**Conclusion** Our work shows that the tumour's spatial architecture is an important factor in adaptive therapy and provides insights into how adaptive therapy leverages both inter- and intra-specific competition to control resistance.

## Plain language summary

Cancer therapy traditionally focuses on maximising tumour cell kill with the aim of achieving a cure, but such aggressive treatment can open up space for drug-resistant cells to grow. In contrast, adaptive therapy aims to leverage competition between drug-sensitive and resistant cells by adjusting treatment to maintain the tumour at a tolerable size, whilst preserving drug-sensitive cells. This approach is being tested in trials but is not yet widely used as deeper understanding of cell-cell competition is required. Here, we used a mathematical model to investigate how strongly, and with whom, resistant cells compete during continuous and adaptive therapy, and applied our insights to hormone therapy in prostate cancer where adaptive therapy has recently been successfully trialed. Our results provide new insights into how adaptive therapy works and show that, by shaping cell competition, the tumour's spatial architecture is important in determining therapy response.

[1] Department of Integrated Mathematical Oncology, H. Lee Moffitt Cancer Center & Research Institute, Tampa, FL, USA. [2] Wolfson Centre for Mathematical Biology, University of Oxford, Oxford, UK. [3]These authors jointly supervised this work: Philip K. Maini, Alexander R. A. Anderson. ✉email: maximilian.strobl@gmail.com; alexander.anderson@moffitt.org

"Can insects become resistant to sprays?"—this is the question entomologist Axel Melander raised in an article of the same title in 1914[1]. At a site in Clarkston, WA, Melander had observed that over 90% of an insect pest called the "San Jose scale" was surviving despite being sprayed with sulphur-lime insecticide[1]. If we were to ask the same question in cancer treatment today we would be met with an equally resounding "yes": while for most cancers it is possible to achieve an initial, possibly significant, burden reduction, many patients recur with drug-resistant disease, or even progress while still under treatment. Drug resistance can develop in a number of ways, including genetic mutations that alter drug binding, changes in gene expression, which activate alternative signalling pathways, or environmentally mediated resistance[2–4]. In the clinic, the main strategy for managing cancer drug resistance is to switch treatment with the aim of finding an agent to which the tumour is still susceptible[3,4]. Similarly, Melander suggested it might be possible to tackle sulphur-lime resistance by switching to oil-based sprays[1]. However, he also foresaw the possibility of, and the challenges arising from, multi-drug resistance[1]. As an alternative, Melander proposed that it might be possible to maintain insecticide sensitivity through less aggressive spraying as this would promote inter-breeding of sensitive and resistant populations and, thus, dilute the resistant genotype[1]. Allocation of spray-free "refuge" patches in the neighbourhood of plots in which an insecticide is used is one modality of modern pest management and is even required by law for the use of certain agents in the US (e.g., Bt-crops[5]). Similarly, research into antibiotic resistance has investigated strategic modulation and combination of treatments to suppress, and ideally reverse, resistance evolution (see Baym et al.[6] for a comprehensive review). For example, Abel zur Wiesch et al.[7] found in a meta-analysis that "adjusted cycling" of drugs, where treatments were switched when resistance was detected, reduced the evolution of antibiotic resistance in hospital wards. Moreover, Hansen et al.[8] have shown that by maintaining drug-sensitive bacteria they can slow the emergence of resistant cells in a bioreactor.

Recently, the concept that treatment de-escalation can delay the emergence of resistance has found application also in oncology. Standard-of-care cancer treatment regimens aim to maximise cell kill through application of the maximum tolerated dose (MTD), in order to achieve a cure. In contrast, an emerging approach called adaptive therapy proposes to focus not on burden reduction, but on burden control in settings, such as advanced, metastatic disease, in which cures are unlikely[9–12]. Eradication strategies free surviving cells from intra-tumoral resource competition, which would otherwise inhibit resistance growth. Adaptive therapy aims to leverage this competition by maintaining drug-sensitive cells in order to avoid, or at least delay, the emergence of resistance[9,11]. A number of pre-clinical studies have demonstrated the feasibility of this approach in ovarian[10], breast[13], colorectal[14], and skin[15] cancer. While large-scale, randomised clinical trials are outstanding, a pilot trial of adaptive therapy in metastatic castration-resistant prostate cancer achieved not only an at least 10 month increase in median time to progression (TTP), but also a 53% reduction in cumulative drug usage in comparison to a contemporaneous control cohort[16]. Further clinical trials in castration-sensitive prostate cancer and melanoma are ongoing (clinicaltrials.gov identifiers NCT03511196 and NCT03543969, respectively).

In addition to testing its feasibility, there has been significant interest in characterising the underpinning eco-evolutionary principles of adaptive therapy through mathematical modelling. We identify three key results. The first insight was derived from approaches which represent the tumour as a mixture of drug-sensitive and resistant cells modelled as a system of two or more ordinary differential equations (ODEs) with competition described by the Lotka–Volterra model from ecology[10,14,15,17–20] or by a matrix game[21]. These analyses have demonstrated that less aggressive treatment allows for longer tumour control under a range of assumptions on the tumour growth law (exponential:[14,17,19,21]; logistic:[10,15,17,19]; Gompertzian:[17–19]; dynamic carrying capacity:[14,20]), and the origin of resistance (pre-existing:[10,14,17–19,21]; acquired[15,18,19,22]; cancer stem-cell-based:[20]). Furthermore, this work predicts that adaptive therapy will be most effective in cases where cures are unlikely due to pre-existing resistance and where at the same time conditions (resistance fraction, proximity to carrying capacity) are such that inter-specific competition with drug-sensitive cells is strong (see ref. [19] for a comprehensive and formal summary of these results). The second key result is that while these conclusions broadly transfer to more complex, spatially explicit tumour models, the strength of spatial constraints on resistant cell growth is important[14,23]. Bacevic et al.[14] showed in a two-dimensional (2-D), on-lattice, agent-based model (ABM) of a tumour spheroid that longer control is achieved if resistance arises in the centre of the tumour compared to when it arises on the edge. Gallaher et al.[23] corroborated this result in a 2-D, off-lattice setting with resistance modelled as a continuum, and further demonstrated that tumour control was adversely affected by high cell motility and cell plasticity. Thirdly, models focussed on metastatic prostate cancer have illustrated how these concepts may be realised in a specific disease pathology[16,20] and how we may enhance tumour control by using a multi-drug approach[20,24,25].

But, what does the competitive landscape of a resistant cell actually look like? With whom do resistant cells compete and at what rate? Even though competition is a key ingredient of adaptive therapy, to the best of our knowledge, no study to-date has explicitly quantified it. Moreover, non-spatial work typically models competition phenomenologically, using Lotka–Volterra dynamics (or a Gompertzian analogue[18,19]). However, this assumes perfect mixing of cells, and is likely an inaccurate description of the dynamics in solid tumours. We, and others[26–29], have recently shown that spatial constraints may alter the nature of tumour evolution away from what may be expected from non-spatial models, an observation which has also been made in the antibiotic resistance community[30,31]. Better understanding the impact of space on the ecology and evolution during treatment is therefore important, and may help to more accurately identify for whom and how to adapt treatment in oncology, and beyond.

The goal of this paper is to study competitive suppression of resistant cells during adaptive therapy and how this is modulated by space. To do so, we developed a simple, 2-D, on-lattice, ABM in which the tumour is assumed to be composed of two cell types: drug-sensitive and resistant cells. The individual-based, spatially explicit nature of our model allowed us to directly quantify competition for space. Leveraging this, we show, for the first time, the competitive suppression of resistant cells during adaptive therapy, and we discuss how the initial proximity to carrying capacity, the initial resistance fraction, the presence of resistance costs, and the rate of cell turnover alter competition. This analysis not only provides useful validation of current theory, but also highlights a seldomly discussed factor: namely, that resistant cells compete not only with sensitive cells, but also with each other. We show that this observation is important in solid tumours, where mixing of cells is limited, because it implies that the spatial distribution of resistant cells strongly impacts treatment response under adaptive therapy. Subsequently, we discuss the implications of our new insights for the modelling of adaptive therapy using ODE models, by comparing our ABM to its equivalent mean-field ODE approximation, which we studied recently[32]. To conclude, we present an analysis in which we fitted our ABM to publicly available, longitudinal data from 65

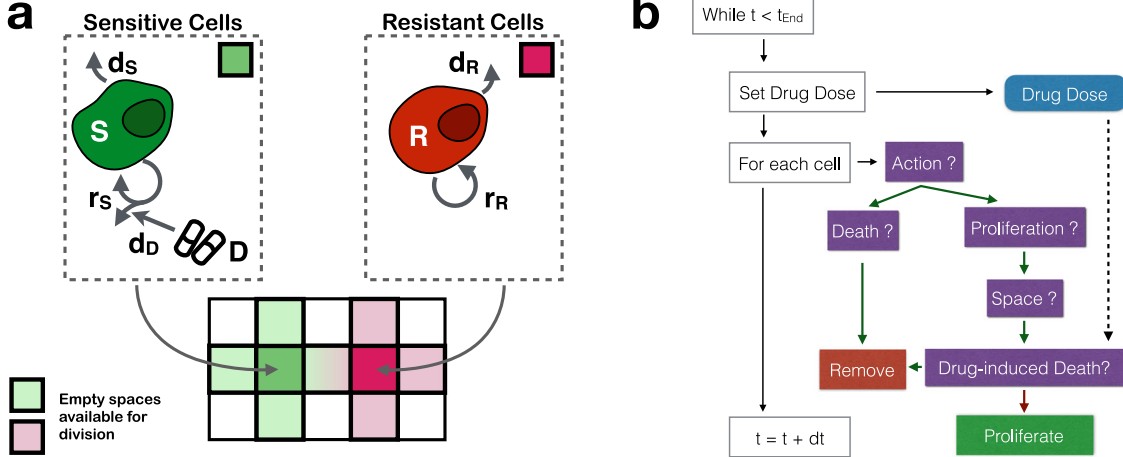

**Fig. 1 The agent-based tumour model. a** The tumour is modelled as a mixture of drug-sensitive ($S$) and resistant cells ($R$), where each cell occupies a square on a 2-D equi-spaced lattice. Cells divide and die at constant rates $r_S$ and $r_R$, and $d_S$ and $d_R$, respectively. Daughter cells are placed into empty squares in a cell's von Neumann neighbourhood. Drug ($D$) will kill dividing sensitive cells at a probability $d_D$. **b** Flow diagram of the simulation algorithm ($t_{End}$: end time of the simulation; dt: simulation time step). For parameter values, see Table 1.

prostate cancer patients undergoing intermittent androgen deprivation therapy. A striking feature of these data is that patients cycle between on- and off-treatment phases at different frequencies. Based on our model fits, we propose that the variations in cycling speed reflect differences in the spatial organisation of these tumours, with implications for the balance of intra- and inter-specific competition between sensitive and resistant cells within. Overall, our work helps to provide a more detailed understanding of spatial competition between sensitive and resistant cells during adaptive therapy and shows that the spatial architecture of the tumour can strongly affect treatment outcomes. While we focus here on cancer, we believe that because of the parallels with strategies explored in other areas, such as antibiotic resistance, our insights may be also of interest to the wider scientific community.

## Methods

**The mathematical model.** Random geno- and phenotypic variation produces tumour cells, which show a degree of drug-resistance even prior to drug exposure. This may manifest as an increased ability to persist and adapt to adverse conditions such as drug exposure, or, though perhaps more rarely, it may take the form of fully developed resistance[3,4]. Selective expansion and further adaptation of this population is thought to be the cause of treatment failure in patients[33,34].

To study the evolutionary dynamics in response to treatment we consider a 2-D, on-lattice, ABM representative of a small region of tumour tissue or a metastatic site. For simplicity we assume that we can divide cells into drug-sensitive or fully drug-resistant subpopulations (Fig. 1a). We choose an on-lattice, agent-based representation as it allows us to explore the role of space and cell-scale stochasticity in a tractable, yet generalisable, way. Each cell occupies a single site in an $l \times l$ square lattice with no-flux boundary conditions, and behaves according to the following rules (Fig. 1):

- Initially, there are a total of $N_0$ cancer cells in the tissue of which a fraction $f_R$ is resistant. Generally, we will assume that the cells are spread randomly throughout the tissue, except in section "The spatial distribution of resistance impacts adaptive therapy by shaping intra-tumoral competition" where we will explore a clustered and a disk-like configuration to dissect the role of the initial conditions more explicitly.

- Sensitive and resistant cells attempt to divide at constant rates $r_S$ and $r_R$ (in units: d$^{-1}$), respectively. If there is at least one empty site in the cell's von Neumann neighbourhood (consisting of the four lattice neighbours, east, west, north, and south, of the cell), then the cell will divide and the daughter will be placed randomly in one of the empty sites in the neighbourhood.

- Cells die at a constant rate $\delta_T$ (in units: d$^{-1}$). For notational convenience, we will express this rate relative to the sensitive cell proliferation rate, so that $d_T = \delta_T / r_S$. Note that this definition of turnover compares the cell death rate to the cells' intrinsic proliferation rate, and is thus not the same as the sometimes measured "cell loss" rate (refs. [35,36]; see also ref. [32] for a further comparison of the two). In addition, we will make the simplifying assumption that that both sensitive and resistant cells die at the same rate, $d_S = d_R = d_T$.

- Movement of cells is neglected.

- The domain is sufficiently small so that drug concentration $D(t) \in [0, D_{Max}]$ is assumed to be spatially homogeneous throughout the tissue, where $D_{Max}$ is the MTD.

- A sensitive cell which is currently undergoing mitosis—that is, it has attempted division and has space available in its neighbourhood—is killed by drug with probability $d_D D(t)$, where $0 \le d_D D_{Max} \le 1$.

- Dead cells are immediately removed from the domain.

We denote the number of cells in each population at time $t$ by $S(t)$ and $R(t)$, and the total number by $N(t) = S(t) + R(t)$, respectively (Table 1).

We consider two treatment schedules:

- Continuous therapy at MTD: $D(t) = D_{Max} \forall t$.

- Adaptive therapy as implemented in the Zhang et al.[16] prostate cancer clinical trial: Treatment is withdrawn once a 50% decrease from the initial tumour size is achieved, and is reinstated if the original tumour size ($N_0$) is reached:

$$D(t) = \begin{cases} D_{Max}, & \text{until } N(t) < 50\% N_0 \\ 0, & \text{until } N(t) = N_0. \end{cases} \quad (1)$$

This results in cycles of on- and off-treatment periods, which maintain the tumour burden at at least 50% its original level for as long as possible, and thereby seek to slow the expansion of resistance.

**Table 1 Summary of mathematical model variables and parameters.**

| Parameter | Description | Value | Comment |
|---|---|---|---|
| $l$ | Grid size (Total number of sites: $l \times l$) | 100 | |
| $dt$ | Simulation time step | 1d | |
| $S(t)$ | Number of sensitive cells | $0–l^2$ | |
| $R(t)$ | Number of resistant cells | $0–l^2$ | |
| $N(t)$ | Total tumour cell number | $0–l^2$ | |
| $r_S$ | Sensitive cell proliferation rate | $0.027\,\mathrm{d}^{-1}$ | Adopted from ref. [16]. |
| $c_R$ | Resistance cost (resistant proliferation rate: $r_R = (1 - c_R)$ $r_S$) | 0–50% | Lower limit:adopted from ref. [23]; Upper limit: assumption of no cost |
| $d_T$ | Cell death rate (relative to $r_S$) | 0–50% | Lower limit: assumption of no turnover; Upper limit: see discussion in ref. [32]. |
| $d_D$ | Drug-induced cell kill probability of sensitive cells at $D(t) = D_{\mathrm{Max}}$ | 0.75 | Adopted from ref. [24]; verified to provide a good fit to prostate cancer in ref. [32]. |
| $n_0 = \frac{N_0}{l^2}$ | Initial cell density (as a percentage of total carrying capacity) | 25–75% | Values within this range reported by ref. [77]. |
| $f_R := \frac{R_0}{N_0}$ | Initial resistant cell fraction (as a percentage of initial cell density) | 0.1–10% | Values within this range reported by ref. [78]. |
| $f_S := \frac{S_0}{N_0}$ | Initial sensitive cell fraction (as a percentage of initial cell density) | 90–99.9% | Determined by $1 - f_R$. |

Progression was determined as a 20% increase from the pre-treatment baseline. However, occasionally it could happen that during adaptive therapy the tumour burden briefly exceeded this target at the end of the first or second off-cycle, due to rapid regrowth of the sensitive cells. As the tumour in these cases was immediately brought back under control upon treatment re-administration, and the focus of our study was progression driven by drug-resistant cells, we neglected these events and only considered a tumour to have progressed if at least 150 days had passed since the start of treatment. This criterion was applied to both adaptive and continuous therapy.

A flow-chart of our model is shown in Fig. 1b, and further implementation details are given in Supplementary Methods 1. We checked convergence (not shown), and performed a consistency analysis[37,38]. This showed that a sample size, $n$, upward of 250 provides a representative sample size for our stochastic simulation algorithm (see Supplementary Fig. 1 and Supplementary Methods 2 for details). The model is implemented in Java 1.8. using the Hybrid Automata Library[39]. Data analysis was carried out in Python 3.6, using Pandas 0.23.4, Matplotlib 2.2.3, Seaborn 0.9.0, and openCV 3.4.9. The time-evolution of the resistant cells' neighbourhood composition was visualised using EvoFreq[40] in R 4.0.2. All code is available on GitHub at https://github.com/MathOnco/strobl2021_space_modulates_competition_AT (see also ref. [41]).

**Comparison with the non-spatial model**. To understand the impact of space we compared the ABM to the following ODE model, which we have studied previously in ref. [32]:

$$\frac{dS}{dt} = r_S \left(1 - \frac{S+R}{K}\right)\left(1 - \frac{2d_D}{D_{\mathrm{Max}}} D(t)\right) S - \delta_T S, \quad (2)$$

$$\frac{dR}{dt} = r_R \left(1 - \frac{S+R}{K}\right) R - \delta_T R, \quad (3)$$

$$N(t) = S(t) + R(t), \quad (4)$$

where $K$ is the carrying capacity, and the initial conditions are given by $S(0) = S_0$, $R(0) = R_0$, and $N_0 = S_0 + R_0$, respectively. We set $K = l^2$ and used the same parameter values as for the agent-based simulation otherwise (Table 1). The equations were solved using the RK45 (used when comparing the ABM and

ODE model in section "How competition is modelled matters") or DOP853 (used for faster computational performance when fitting the patient data in section "The cycling frequency of patients undergoing intermittent androgen deprivation therapy may reflect different spatial distributions of resistance") explicit Runge–Kutta schemes provided in Scipy (for further details see ref. [32]).

**Model parameters**. We parametrised our model using values from the literature for prostate cancer (Table 1). We want to stress, however, that the aim of our work was to develop qualitative understanding, not to make quantitative predictions directed at prostate cancer. As such, our predictions should be interpreted not in a quantitative ("treatment X will achieve a TTP of Y months"), but in a qualitative fashion ("treatment X will achieve a longer TTP than treatment Y because of mechanism Z").

**Analysis of patient data**. In order to examine whether our model could explain differences in the cycling speed of patients undergoing intermittent androgen deprivation therapy, we fitted it to the publicly available, longitudinal response data from the Phase II trial by Bruchovsky et al.[42]. The data were downloaded from http://www.nicholasbruchovsky.com/clinicalResearch.html in July 2020. So as to avoid potentially confounding effects from a change in the number of lesions, patients who developed a metastasis were excluded from analysis. Furthermore, we decided to exclude two further patients (Patients 2 & 104) from our analysis because their prostate-specific antigen (PSA) dynamics were inconsistent with the reported treatment schedules. These patients display oscillating PSA values indicative of treatment cycling but there are no reported changes in treatment, suggesting there may have been a mistake in the data reporting in these cases (see also Supplementary Fig. 2). This yielded data from a total of 65 patients. The model was fitted to the normalised PSA measurements by minimising the root mean-squared error between the data and the predictions (normalisation relative to PSA level at start of treatment). Given the stochastic nature of the ABM, each candidate fit was assembled from 25 independent stochastic replicates. Optimisation was carried out using the basin-hopping algorithm in Scipy[43] employing default search parameters and a maximum of either 50 (when fitting 2 parameters) or 75

optimisation steps (when fitting 4 parameters). To escape potential local minima, optimisation was repeated 10 times for each patient from different, randomly chosen initial conditions, with only the best fit according to the Akaike Information Criterion (AIC) taken forward for analysis. We fitted the model in three different ways: (i) each of $n_0$, $f_R$, $c_R$, and $d_T$ was allowed to be a patient-specific parameter (which we term "Model 1"), (ii) only $n_0$ and $f_R$ were allowed to be patient-specific with $c_R$ and $d_T$ fixed at the mean values obtained in (i) ($c_R = 0.78$, $d_T = 0.14$; Model 2), and (iii) $c_R$ and $d_T$ were allowed to be patient-specific with $n_0$ and $f_R$ fixed at the mean values obtained in (i) ($n_0 = 0.59$, $f_R = 0.04$; Model 3). For 8 patients, Model 1 failed to recapitulate the cycling nature of the patients' trajectories, yielding simple straight lines instead (e.g., Patient 52 in Supplementary Fig. 2). Owing to this discrepancy it was unclear that the associated parameter values would be representative of the tumour biology, and so we excluded these patients when computing the mean of the parameters taken forward to Models 2 and 3. For the same reasons, we excluded these 8 patients and one additional patient (Patient 13) when assessing the correlation between cycling speed, and the cost and turnover estimates of Model 3. This analysis was thus based on the data from 56 patients in total. Excluded patients are marked by a grey background in Supplementary Figs. 2 & 3, respectively. Classification of patients into "progressing" and "non-progressing" was taken from the annotation provided in the data, where progression is defined as a series of three sequential increases of serum PSA > 4.0 μg/L despite castrate levels of serum testosterone. Overviews of all fits of Models 1 and 3 are shown in Supplementary Figs. 2 & 3, respectively. Fitting was done using the lmfit package in Python[44] (version 1.0.1.). As this was a retrospective analysis of a study that was previously approved by the institutional review boards/independent ethics committees of each participating site, and whose data were available in the public domain, no further ethical approval or informed consent was required.

**Reporting summary**. Further information on research design is available in the Nature Research Reporting Summary linked to this article.

## Results

The aim of adaptive therapy is to delay disease progression by leveraging intra-tumoral competition. However, how frequently is a resistant cell actually inhibited from division, and by what? And how does this compare to continuous treatment? To address these questions, we studied matched simulations in which the same tumour (identical parameters and random number seed) was treated once with continuous treatment and once adaptively, using the 50% rule employed in the clinical trial by Zhang et al.[16] (Equation (1)). To quantify the benefit of each protocol we adopted the progression criterion used in a number of previous studies (e.g., refs. [19,23,32]), which is based on a modified version of the RECIST criteria[45], and which determines progression as a 20% increase in tumour cell number relative to the start of treatment.

**Demonstration of competitive suppression of resistance during adaptive therapy**. To gain intuition, we first considered a simple example. We seeded cells randomly to fill 75% of the domain and assumed that 0.1% of these cells were resistant prior to treatment. For simplicity, we assumed that there was no cell death, and no resistance costs. Despite this lack of resistant costs and turnover, we find that adaptive therapy provides the superior resistance control, extending TTP by about 15.3 months (95% CI: [14.81m, 15.86m], Fig. 2a). Examination of the spatial dynamics in the

simulation clearly illustrates how resistant colonies are, at least temporarily, kept in check by neighbouring sensitive cell patches (Fig. 2b and Supplementary Movie 1). At the same time we note that due to the stochastic nature of the simulations, resulting, for example, in different initial spatial distributions of the cells, there is variation in how many adaptive therapy cycles are completed and how long progression can be delayed, despite using identical parameters (Supplementary Fig. 4). This indicates that stochastic effects may also play a role in determining outcome, an observation we will return to later in this paper.

While competition is thought to be the driving mechanism behind adaptive therapy, it is challenging to assert its role in real tumours. This is because it is difficult to rule out the possibility that the benefit of adaptive therapy may be due to other confounding factors, such as the effect of treatment de-escalation on tumour vasculature or the immune response[13,46,47]. However, in our computational model we can directly monitor the interaction between cells, as well as rule out such confounding factors. We seized this opportunity to explicitly measure how often a resistant cell is blocked from division in our simulations (by any cell). Our results reveal a rapid and noticeable intensification of growth inhibition whenever treatment is withdrawn during adaptive therapy ("spikes" in Fig. 2c; left panel). This results in a level of growth inhibition, which is consistently higher than that during continuous therapy until late in the course of treatment, and demonstrates how adaptive therapy leverages inter-specific competition.

**Not only inter- but also intra-specific competition is a factor in adaptive therapy**. That being said, Fig. 2c also shows that the competitive suppression of resistance generally increases over time, and, importantly, that this holds true not only under adaptive, but also under continuous, therapy. To investigate why this is the case, we analysed which cells are responsible for the suppression, by computing the average composition of a resistant cell's neighbourhood in the simulations. We find that the most frequent competitor of a resistant cell under either protocol is not a sensitive cell, but other resistant cells (Fig. 2d). In fact, Fig. 2d shows that inter-specific competition with sensitive cells, which is responsible for the large benefit of adaptive therapy, is experienced only by a small proportion of resistant cells. This is because as the resistant colonies become larger, most resistant cells become trapped in the core and are blocked by other resistant cells at the edge of the colony (Fig. 2b). During adaptive therapy sensitive cells block growth at the edge of resistant colonies which, in turn, also inhibits resistant cells in the centre (Fig. 2b). We conclude that in solid tumours, in which there is only a limited degree of mixing of cells, not only inter-specific but also intra-specific competition may play an important role in adaptive therapy.

**High tumour cell density and low initial resistance fraction maximise competition from sensitive cells**. Previous work by our group and others[18,19,23,32,48] has shown that the closer a tumour is initially to its carrying capacity ($n_0$), and the smaller the initial resistance fraction ($f_R$), the greater will be the gain in TTP by adaptive therapy. A parameter sweep confirms that this is also the case in the current model (Fig. 2e). To better understand why, we studied how these two factors impact the competitive suppression of resistant cells. We find that $n_0$ and $f_R$ have distinct effects on the number and impact of each adaptive therapy cycle, where we define a cycle as the time period between two sequential crossings of the baseline tumour size, consisting of an on- and an off-treatment period. Specifically, we observe that when we reduce $n_0$, treatment still goes through almost the same number of adaptive therapy cycles, but the impact of each cycle on the

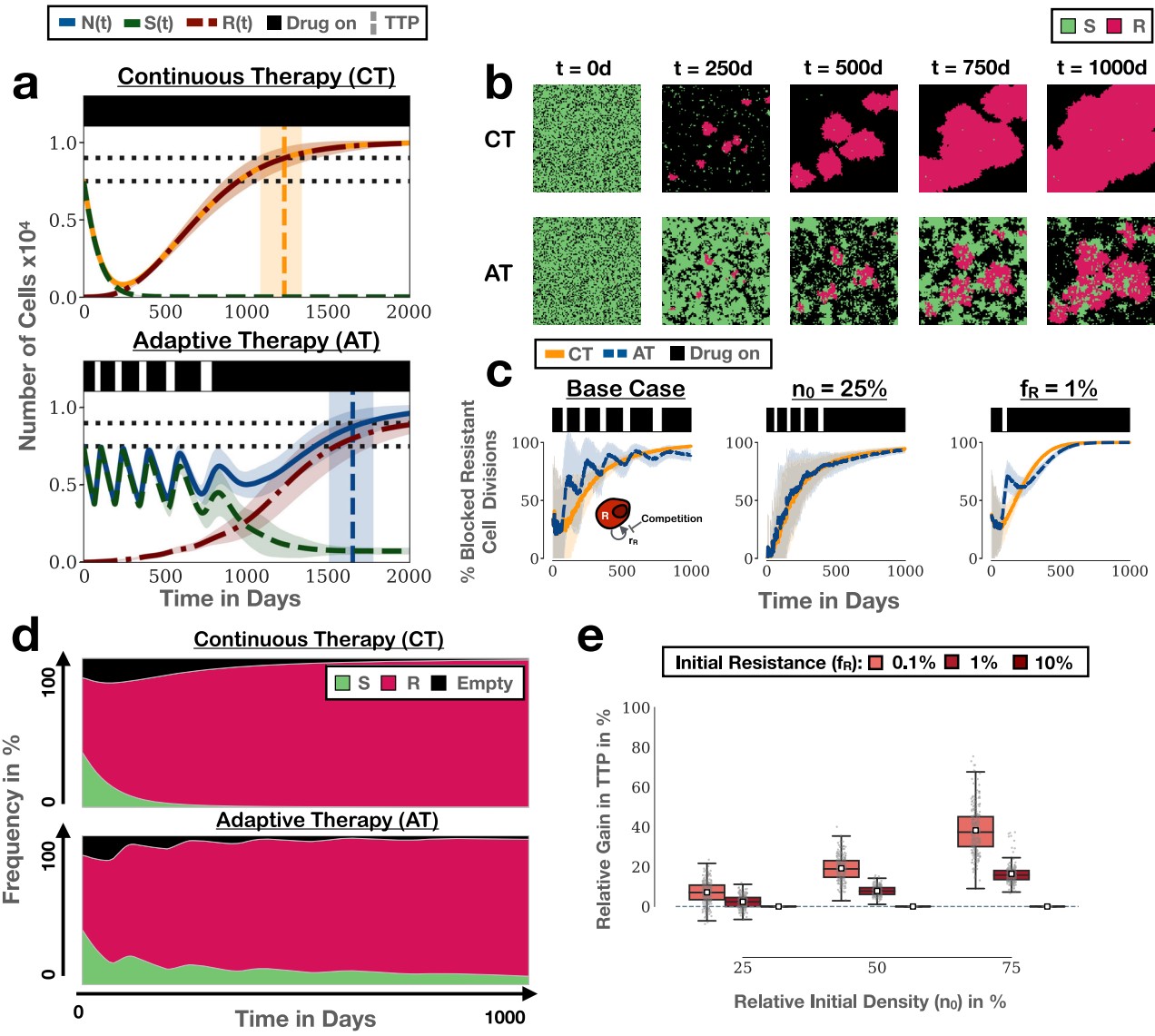

**Fig. 2 Adaptive therapy prolongs TTP by leveraging inter- and intra-tumoral competition.** AT: adaptive therapy, CT: continuous therapy, TTP: time to progression, ABM: agent-based model. **a** Example simulations of the treatment dynamics under continuous and adaptive therapy in our ABM ($(n_0, f_R) = (75\%, 0.1\%)$). Shown are the mean and standard deviation (shading) of the tumour cell number (sensitive, resistant and total; $n = 250$ independent replicates). Black bars represent a typical treatment schedule (see also Supplementary Fig. 4). Horizontal dotted lines show the initial cell number and the cell number at progression, respectively. Vertical lines and associated shading mark the mean, and the 1st and 3rd quartile of the distribution of TTP. **b** Snapshots from one of the replicates in **a** illustrating how adaptive therapy delays competitive release of resistance. **c** Adaptive therapy (blue line) results in a higher proportion of blocked resistant cell divisions than continuous therapy (yellow line) in the initial stages of treatment in the simulations in **a** (left panel). However, the ability of adaptive therapy to induce blocking depends on the initial tumour cell density (centre panel; $(n_0, f_R)$ = (25%, 0.1%)) and the initial resistance fraction (right panel; $(n_0, f_R)$ = (75%, 1%)). Lines and shading indicate mean and standard deviation, respectively. **d** Average frequency of sensitive cells, resistant cells, or empty space in a resistant cell's neighbourhood in the simulations in **a**. This shows that there is not only inter-, but also significant, intra-specific competition during adaptive therapy. Values are the mean in the von Neumann neighbourhood across all resistant cells from 250 independent simulations. **e** Gain in TTP achieved by adaptive therapy relative to continuous therapy as a function of $n_0$ and $f_R$ ($n = 1000$ independent replicates per condition). Adaptive therapy yields the greatest benefit when tumours are close to their carrying capacity and resistance is rare. The box, centre line, small white squares, and whiskers denote the inter-quartile range, median, mean, and 1.5x inter-quartile range, respectively. Negative values denote cases in which continuous therapy achieves a longer TTP. Resistance costs and turnover are assumed to be 0 throughout this figure.

reduction of the resistant cell growth rate is greatly diminished (Fig. 2c; middle panel). Owing to the low cell density there is effectively no inter-specific competition (although there is still noticeable intra-specific competition as can be seen by the general upwards trend in Fig. 2c).

Conversely, if we increase the initial resistance fraction, but keep cell density high ($n_0 = 75\%$), then treatment fails after only a single cycle. However, this single cycle still induces blocking of resistant cells and results in a benefit in TTP (Fig. 2c; right panel). We also note that the competitive suppression of resistant cells generally increases more rapidly in this case compared to when there were fewer resistant cells present, reflecting stronger intra-specific competition (compare right and left panels in Fig. 2c). One implication of these observations is that tumours with a

higher resistance fraction may require an even further de-escalation of treatment to maintain sensitive cells for as long as possible. Indeed, treatment of the same tumour with an adaptive protocol in which drug is withdrawn at a 30% reduction in size, instead of 50%, increases the time gained from 2.5 months to 6.5 months (Supplementary Fig. 5). We conclude that close proximity to carrying capacity and small initial resistance fractions are two factors that help to maximise the impact of inter-specific competition.

**Resistance costs slow progression but in the absence of turn-over their impact on competitive suppression is small**. Having examined how the initial tumour composition impacts competition, we next investigated the influence of the cell kinetic parameters, specifically the resistant cell proliferation rate and the cell turnover rate. The concept of a "cost of resistance", a decreased proliferation rate in a drug-free environment due to, for example, increased energy consumption, has been an important argument in the original development of adaptive therapy[9–11]. While such fitness costs have been reported, for example, in breast cancer[23], others have found resistant cells pay no cost[15], or even grow faster than sensitive cells[49], as a result of which the role of resistance costs in adaptive therapy has been a controversial topic.

Using a minimalistic, sensitive/resistant Lotka–Volterra ODE model we recently showed that the rate of cell turnover is an important factor to consider when evaluating the impact of resistance costs[32]. To corroborate this insight, and to better understand how these factors influence competition, we carried out simulations with or without costs or turnover (Fig. 3a). We find that the addition of a resistance cost on its own slows down progression of the tumour under both protocols (Fig. 3a), visible as a reduction in the size of resistant colonies (Fig. 3b; see also Supplementary Movie 2). However, an increased benefit of adaptive therapy is seen only when the resistant fraction ($f_R$) is small, and this increase is modest (Fig. 3a; right panel). Mapping out the impact of resistance costs in more detail corroborates this result (Fig. 3c; left panel). Consistent with these observations, we find that the profile of competitive suppression over time changes relatively little when a resistant cost is added (Fig. 3d). Suppression increases more slowly but continues to follow the same trajectory, with only a small difference between the two protocols.

**Turnover amplifies the impact of competitive suppression**. Next, we repeated this analysis whilst assuming a cell turnover rate of 30% relative to the proliferation rate, which is the value we previously estimated for prostate cancer[32]. For simplicity, we assumed the same death rate for both sensitive and resistant cells and that dead cells are immediately removed from the domain. We find that the inclusion of cell death increases the average number of adaptive therapy cycles, and with it the benefit of adaptive therapy (Fig. 3a). Moreover, consistent with our prior ODE study[32], we observe that turnover amplifies the effect of resistance costs (Fig. 3a, c).

To explain why turnover facilitates the control of the drug-resistant population, we examined its impact on the competitive suppression experienced by resistant cells. Interestingly, and somewhat counter-intuitively, we find that turnover reduces blocking of resistant cell divisions (Fig. 3d). This is because turnover frees up space for cell division (see gaps at the centre of colonies in Fig. 3b; see also Supplementary Movie 2).

So, why does the benefit of adaptive therapy increase? To explain this, we leverage an argument first proposed by Hansen et al.[48], which states that because we have complete control over the sensitive population (we can reduce their size at will), TTP is entirely driven by the net-growth rate of the resistant population,

i.e. the balance between birth and death (Fig. 3e). Computing the net-growth rates in our simulations, we see that even though cell proliferation is less restricted with turnover, the net-growth rate is still reduced (Fig. 3f). This has two effects: Firstly, drug can be withdrawn more often during the course of adaptive treatment, and secondly, the impact of the blocking that does occur is amplified (spikes during treatment withdrawal reach smaller net-growth values). In summary, these results corroborate our hypothesis that the rate of tumour cell death is an important factor in adaptive therapy, and show how it modulates the impact of competitive suppression.

**The spatial distribution of resistance impacts adaptive therapy by shaping intra-tumoral competition**. In our analyses up to this point we had assumed that resistant cells were seeded randomly in the domain, so that the resistant population emerges from multiple, independent colonies (or "nests") simultaneously. This was so as to mimic the diffuse structure of invasive cancers in which tumour islets are interspersed by areas of tumour stroma, necrosis, or the remnants of the normal tissue. However, given our insights on the role of intra-specific competition, and the prior results by Bacevic et al.[14] and Gallaher et al.[23] regarding the importance of spatial constraints, we hypothesised that the initial spatial distribution of resistance may be an important factor in adaptive therapy. To investigate this we conducted a series of experiments in which we seeded the same number of resistant cells, either as a single cluster, or as a set of two clusters, at varying distances apart (Fig. 4a; top row). We placed eight resistant cells as a $2 \times 4$ rectangle in the centre of the domain. Subsequently, we compared this scenario to those in which we split this cluster into two nests of size $2 \times 2$, which we placed at varying distances apart from each other. Sensitive cells were seeded randomly in the domain to achieve a total initial density of $n_0 = 50\%$.

Our results confirm that the speed of progression is determined, not only by the initial number of resistant cells, but also by their distribution. For example, even though all three simulations in Fig. 4a started from eight resistant cells, the further apart the two nests, the more resistant cells are present at 700 days. Computing TTP confirms this (Fig. 4b). Moreover, with increasing separation the benefit of adaptive therapy also declines (Fig. 4c).

This observation has an important implication. Based on the hypothesis that adaptive therapy is driven by inter-specific competition between sensitive and resistant cells, we would have expected the opposite to hold true, as placing the nests apart maximises the opportunity for interaction between the cell populations (Fig. 4a). As such, these new findings corroborate the notion that not only inter- but also intra-specific competition plays a role in adaptive therapy. This is because competition with sensitive cells is, in a sense, a double-edged sword. We can control sensitive cells with treatment, and in the absence of treatment they may even have a competitive advantage over resistant cells. However, in the presence of treatment this advantage is lost, and by clearing sensitive cells we, in fact, open up space allowing resistant cells to expand. In contrast, when resistant cells grow adjacent to other resistant cells, any successful division is a zero-sum game, as it simply replaces a resistant cell, which was previously in this position. As such, while there is more interaction with sensitive cells when the resistant nests are seeded apart, better control is achieved when they are clustered together because it allows adaptive therapy to leverage both inter- as well as intra-specific competition.

Another reason why adaptive therapy is less effective when there are two separate nests, rather than one, is illustrated in Fig. 4d. While the left of the two nests is initially constrained by sensitive cells ($t = 250$d), the right nest is not and is able to

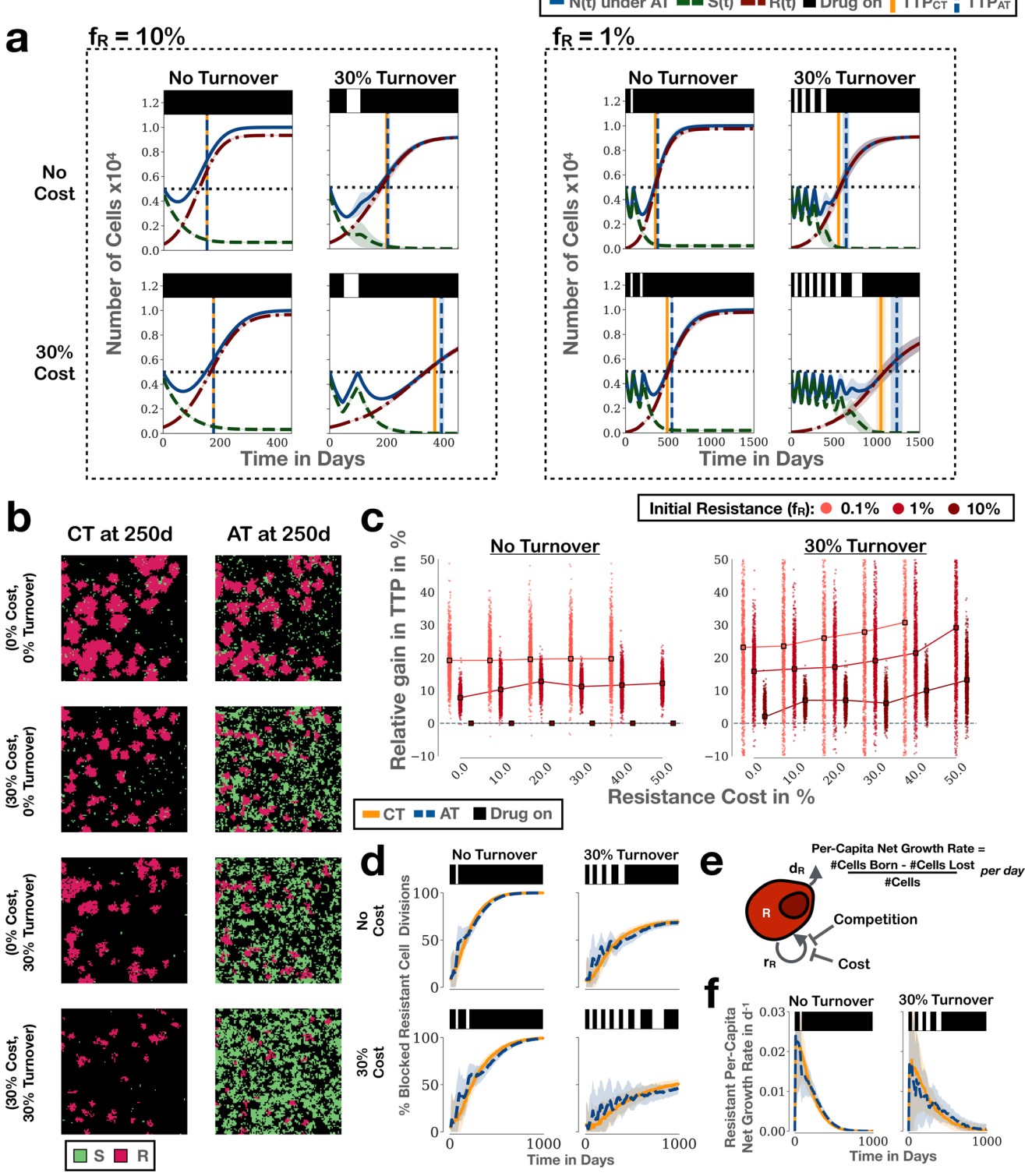

expand ($t = 500$d). This triggers more and more treatment, which eventually results in the competitive release also of the left nest ($t = 1000$d).

Finally, we note that similar considerations also apply to the sensitive cells. As we assume that drug affects only dividing sensitive cells, the strength of intra-specific competition between sensitive cells plays an important role in determining the speed at which a tumour will go through a cycle of adaptive therapy. The

impact of this can be seen, for example, when comparing simulations with different levels of turnover (Fig. 3a), or when we seed the cells as a cluster in the centre of the domain rather than seeding them randomly (Supplementary Discussion 1 and Supplementary Fig. 6). We will return to this point in section "The cycling frequency of patients undergoing intermittent androgen deprivation therapy may reflect different spatial distributions of resistance".

**Fig. 3 The impact of resistance costs and cellular turnover on competitive suppression and adaptive therapy.** AT: adaptive therapy, CT: continuous therapy, TTP: time to progression, TTP$_{CT}$: TTP under CT, TTP$_{AT}$: TTP under AT. **a** Simulations illustrating the role of resistance costs and turnover on the treatment response to adaptive therapy ($n_0 = 50\%$). Vertical lines and associated shading mark the mean, and the 1st and 3rd quartiles of the distribution of TTP. **b** Snapshots at time $t = 250$d from one of the replicates in the right hand panel in **a**) (($n_0, f_R$) = (50%, 1%); see also Supplementary Movie 2). All 8 simulations were started from the same initial conditions and treated either continuously or adaptively. **c** Relative benefit of adaptive therapy over continuous therapy as a function of resistance cost, resistance fraction, and turnover ($n_0 = 50\%$). This illustrates that turnover modulates adaptive therapy and increases the impact of a resistance cost. Points show individual simulations and squares mark the mean ($n = 1000$ independent replicates per condition). **d** Impact of resistance costs and turnover on blocking of resistant cell division for the simulations in the right panel in **a**. **e** TTP is determined by the per-capita net-growth rate of resistant cells, which depends on both the birth and death rate of cells. Importantly, if death rates are high, then even moderate inhibition of cell proliferation by competition may result in large reductions in per-capita growth rate. **f** Per-capita growth rate of resistant cells as a function of time, illustrating how turnover helps to amplify the effects of competition on the resistant population's net-growth rate (($n_0, f_R$) = (50%, 1%)). Unless otherwise stated, lines and shading in this figure denote the mean and standard deviation of $n = 250$ independent replicates, respectively.

**The spatial distribution of resistance modulates the impact of cost and turnover.** Next, we studied how space modifies the impact of resistance costs and turnover. We find that in the presence of turnover the mean TTP and the benefit of adaptive therapy also decrease with increasing nest separation (Fig. 4e). At the same time, turnover can cause the random extinction of one of the two nests, or both, which greatly extends TTP and results in a high variability in outcomes, especially if the separation between nests is large (insets in Fig. 4e). In addition, greater spread of resistant cells reduces the benefit adaptive therapy can derive from resistance cost and turnover (Fig. 4f), while clustering the cells in the centre of the domain increases it (Supplementary Fig. 6). We conclude that the spatial distribution of resistance plays a significant role in the response to adaptive therapy, and its potential benefit.

**How competition is modelled matters.** In the second part of this paper we will discuss implications of our new insights for the mathematical modelling of adaptive therapy, and for the treatment dynamics, which we may observe in patients. Most mathematical models of adaptive therapy to-date have assumed perfect mixing of cells, so that the growth inhibition due to competition is given by the classical Lotka–Volterra competition model from ecology. However, how appropriate is this approximation for solid tumours, in which the rate of mixing is small, due to limited migration and spatial constraints?

To address this question, we compared the ABM with its corresponding Lotka–Volterra ODE model (Equations (2)–(4); see also ref. [32] for an in-depth discussion of this model), assuming the same (exponential) rates for cell division, death and drug kill (Table 1 and Fig. 5a, b). This shows that while the dynamics agree qualitatively, there are important quantitative differences. Firstly, the ABM tends to predict longer TTP under both regimens (Fig. 5a), and this discrepancy increases the higher the initial density and the smaller the initial resistance fraction (Fig. 5b). At the same time, the cycling frequency is higher in the ABM with both short on- as well as off-times, resulting in a larger number of cycles (Fig. 5a). Importantly, when we compare the relative benefit of adaptive therapy predicted by the two models we find that the ABM tends to forecast a smaller gain than the ODE model, especially if turnover is included in the model (Fig. 5c).

To understand why the discrepancy between the ABM and ODE model arises, we examined the growth dynamics in both models in more detail. We find that when the initial resistance fraction is small, the resistant population in the ABM expands more slowly than in the ODE model, but the converse holds true when the initial resistance fraction is large (Supplementary Fig. 7). To explain this, we simulated the resistant population in isolation, starting from different initial cell numbers. Our results demonstrate that different initial cell numbers generate distinct growth kinetics (Fig. 5d). When initiated from two cells, the resistant population expands as two colonies and grows much more slowly than predicted by a logistic ODE model, as most cells are trapped by their neighbours (Fig. 5d). As the number of cells, and so the number of independent nests and the surface to volume ratio, is increased, the growth of the population speeds up until it exceeds that of logistic growth (Fig. 5d). This explains the differences in Fig. 5b, and highlights again the importance of the initial spatial distribution of resistance. Furthermore, it indicates that the Lotka–Volterra model, which assumes logistic growth, will likely be an inaccurate description of intra-tumoral competition in solid tumours.

Similarly, we can explain the reduced impact of turnover on TTP by differences in cell growth dynamics between the two models. Recall that turnover has two effects: On the one hand, it limits a cell's lifespan and so the number of opportunities it has to divide. The higher the turnover, the smaller is this number, and the greater is the impact of a blocked division[32]. On the other hand, death of a neighbour opens up space for cell division, which partially off-sets the cell loss caused by turnover (Fig. 3d). Importantly, in the ABM each cell can divide into four potential sites, which allows for more divisions to take place at high cell density than predicted by the ODE model, which is why the impact of turnover is reduced (Supplementary Fig. 8).

**Stochastic extinction and competition-induced morphology changes can cause adaptive therapy to fail before continuous therapy.** Comparing the two models reveals a further important difference: There is variation in the possible outcomes in the ABM, so that despite identical parameters there can be noticeable differences in the benefits derived from adaptive therapy (Fig. 5c). Moreover, some patients are, in fact, predicted to benefit more from continuous than from adaptive therapy—an outcome not possible in the ODE model (Fig. 5c and see also Fig. 2e; for a proof of the latter see refs. [19,32]). To investigate how frequently, and why, this occurs, we show a breakdown of the possible outcomes for two different initial tumour compositions in Fig. 6a, b. We make two important observations: Firstly, while one of the strongest arguments against adaptive therapy is that it foregoes the chance of a cure, this is not the main reason for inferior results in our simulations. Instead, in the majority of cases with inferior outcome the tumour still progresses, but is controlled for longer by continuous therapy. Secondly, Fig. 6a, b show that variability in outcomes and the likelihood of inferior results increases with cost and turnover and decreases with the number of cells in the simulation. This indicates that these failures are driven by stochastic effects, a conclusion which is further supported by the fact that variability in outcome decreases when we increase the simulation's domain size (Supplementary Fig. 9a). Together, we conclude that the circumstances under which adaptive therapy can achieve the most favourable results on average may also be associated with a great variability in outcomes.

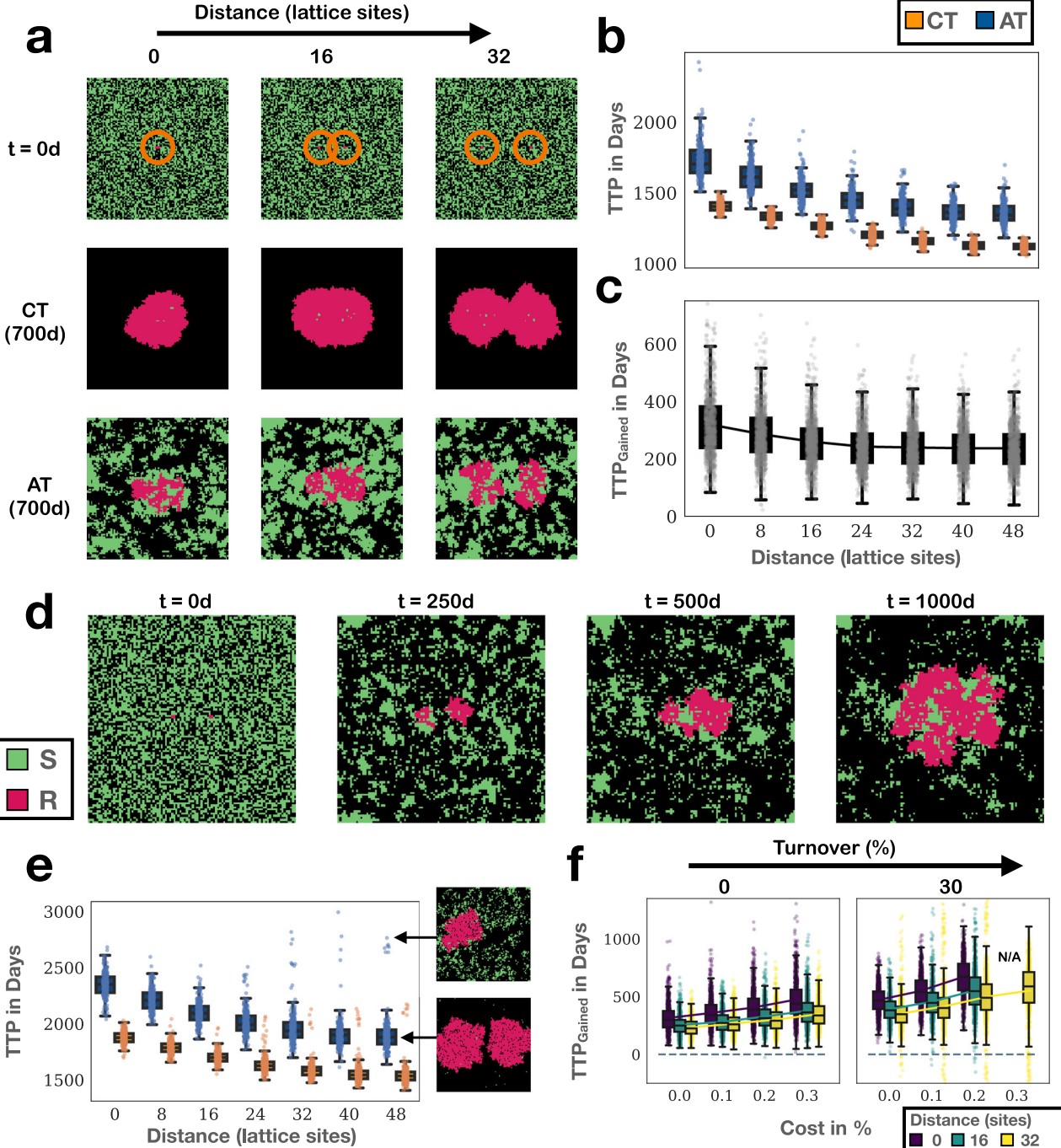

**Fig. 4 The impact of the initial spatial distribution of resistance on adaptive therapy highlights the role of intra-specific competition.** AT: adaptive therapy, CT: continuous therapy, TTP: time to progression. **a** We seeded eight resistant cells either as a cluster in the centre, or as two nests at varying distances apart (marked with orange rings in the top row). Shown are representative snapshots from simulations for three different initial levels of separation $((n_O, c_R, d_T) = (50\%, 0\%, 0\%))$. Observe how there is more scope for interaction between sensitive and resistant cells as nests are seeded further apart, suggesting that these should be better controllable with adaptive therapy. **b** However, TTP of continuous and adaptive therapy decreases as the separation between the nests increases ($n = 1000$ replicates per conditions; parameters as in **a**). **c** Similarly, the benefit of adaptive therapy decreases the greater the separation between the nests ($TTP_{Gained} = TTP_{AT} - TTP_{CT}$, where $TTP_{AT}$ and $TTP_{CT}$ is the time to progression under adaptive and continuous therapy, respectively; $n = 1000$ replicates per conditions; parameters as in **a**). This indicates that not only intra- but also inter-specific competition are important in adaptive therapy. **d** Representative snapshot from a simulation in **a** in which the nests are initially 16 lattice sites apart, which illustrates one reason why control of multiple nests is more challenging. While the left nest can initially be controlled, the right nest escapes and, in turn, triggers release of the left nest. **e** TTP as a function of initial separation distance in the presence of turnover ($d_T = 30\%$; $n = 1000$ replicates per condition). Turnover can cause extinction of one of the two nests, which greatly increases TTP (see insets, which show simulation snapshots at $t = 1500$d). **f** Effect of the initial spatial distribution of resistance on the relationship between cost, turnover and gain of adaptive therapy ($n = 1000$ replicates per condition). "N/A" indicates that data were not shown because less than 1/4 of simulations had completed within 10 years (cured or progressed). Throughout the figure, the box, centre line, and whiskers in box plots denote the inter-quartile range, median, and 1.5x inter-quartile range, respectively.

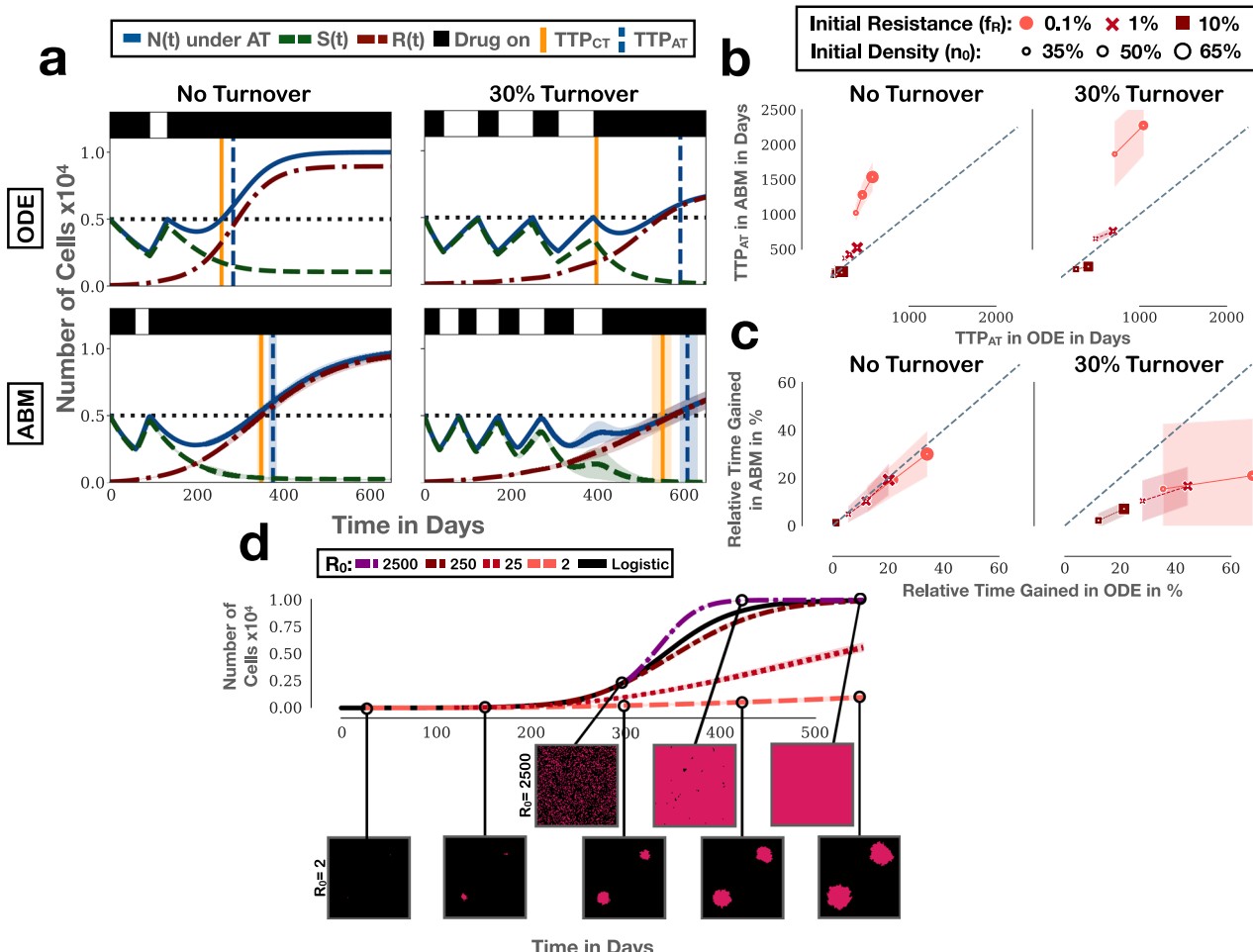

**Fig. 5 Comparison of the spatial ABM with the corresponding non-spatial ODE model from[32], which uses a Lotka–Volterra model to describe competitive growth inhibition (Equations (2)–(4)).** AT: adaptive therapy, CT: continuous therapy, TTP: time to progression, $TTP_{CT}$: TTP under CT, $TTP_{AT}$: TTP under AT, ODE: ordinary differential equation model, ABM: agent-based model. **a** ODE and ABM simulations with the same parameters $((n_0, f_R, c_R) = (50\%, 1\%, 0\%)$; $n = 250$ replicates for the ABM). **b** Comparison of the TTP under adaptive therapy in the ODE and ABM ($c_R = 0\%$). The ABM predicts faster progression when initial resistant cell numbers are high, and slower progression when numbers are low ($n = 1000$ replicates of the ABM). Note that in the presence of turnover, tumours initialised at $n_0 = 65\%$ can not grow above the threshold defining progression and thus, no TTP can be obtained. **c** Comparison of the relative benefit of adaptive therapy ($(TTP_{AT} - TTP_{CT})/TTP_{CT}$) in the two models. The ODE model tends to predict larger benefit than the ABM. Again, no TTP can be obtained for $n_0 = 65\%$. **d** Depending on the initial cell number ($R_O$) the resistant population will grow sub- or super-logistically (logistic growth: black solid line). Resistant cells were seeded and grown in isolation and without drug ($n = 250$). To allow direct comparison with logistic growth, the ABM curves were shifted to start at the time at which the logistic model reached the corresponding starting number of cells. Throughout this figure lines and shading denote the mean and standard deviation, respectively.

While the stochastic nature of the failures makes it difficult to pinpoint exactly why they occur, we identify two patterns, which iterate the importance of the spatial architecture of the tumour. Firstly, random cell death can make individual nests go extinct, which subsequently results in slower progression (Fig. 6c, d). This effect is amplified the further the nests are located apart (recall section "The spatial distribution of resistance impacts adaptive therapy by shaping intra-tumoral competition"), which in the case shown here results in a difference in TTP of over 1800 days (the largest observed among the simulations in Fig. 6a).

However, as seen in Fig. 6a, inferior control under adaptive therapy can occur also in the absence of turnover. To explain why, we show in Fig. 6e, f the simulation in the absence of cost and turnover ($(c_R, d_T) = (0, 0)$) in which adaptive therapy performed most poorly. Looking closely at the morphology of the expanding resistant colonies under both treatment arms we observe that the competitive inhibition by sensitive cells induces a more branching growth pattern in the adaptively treated tumour (insets in Fig. 6f).

This has the effect that once the sensitive cells have been cleared by drug, the larger surface to volume ratio allows these colonies to expand more quickly, which is the reason for the earlier progression under adaptive therapy. To provide more quantitative evidence for this argument, we computed the convexity ("ruggedness") of resistant nests at 250d after treatment initiation, which confirms that more uneven looking nests are correlated with poorer performance of adaptive therapy (Supplementary Fig. 9b). To sum up, when cell numbers are large the results from the ODE model about the non-inferiority of AT holds true, but if resistant cell numbers are small, then the tumour's spatial architecture can amplify stochastic birth/death events and cause significant inter-patient variability in outcome. This may need to be considered in future experimental and trial design.

**The cycling frequency of patients undergoing intermittent androgen deprivation therapy may reflect different spatial**

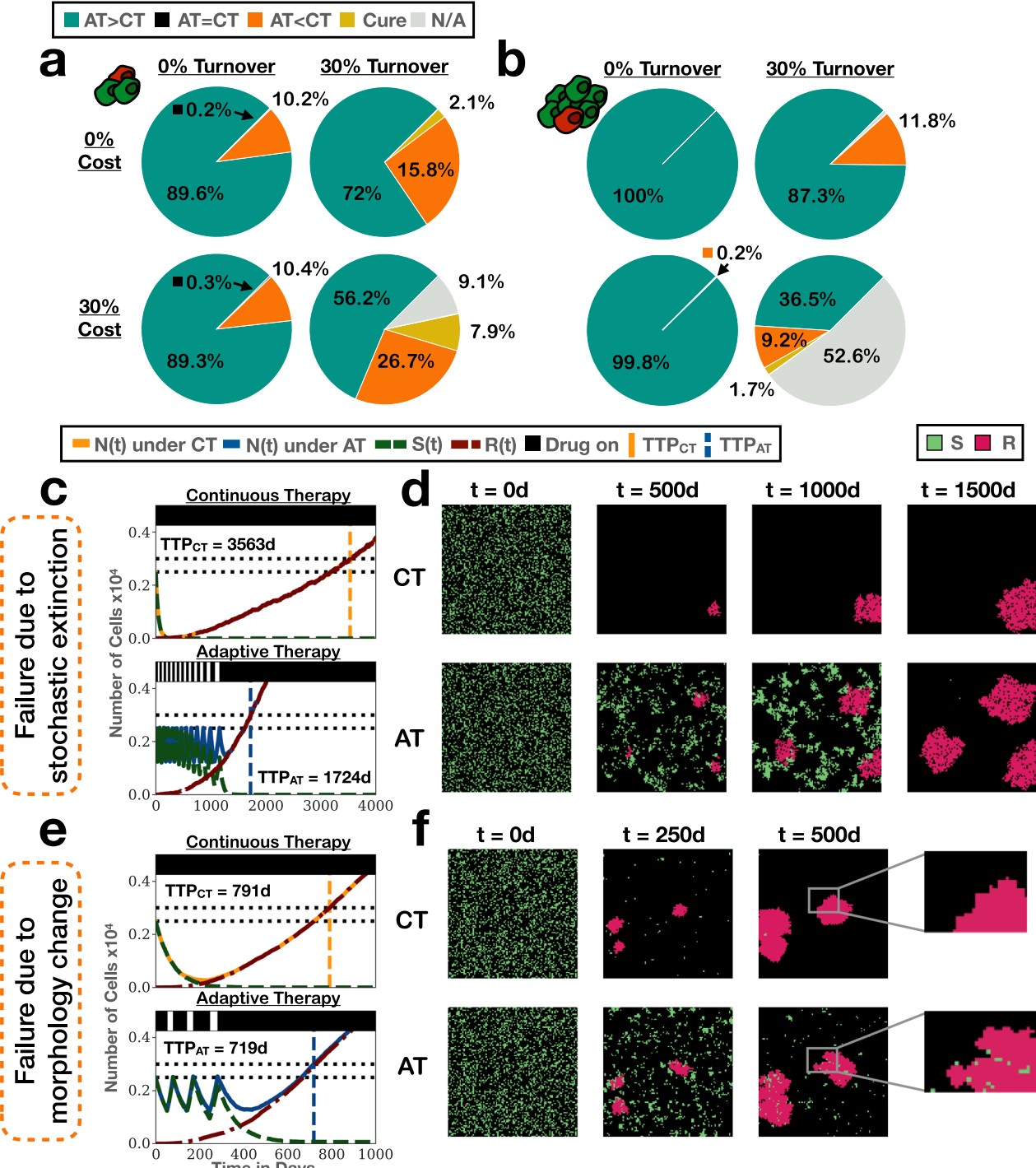

**Fig. 6 While adaptive therapy is beneficial on average, stochastic and spatial effects can result in inferior outcomes in a subgroup of simulations in the ABM.** AT: adaptive therapy, CT: continuous therapy, TTP: time to progression, $TTP_{CT}$: TTP under CT, $TTP_{AT}$: TTP under AT, ABM: agent-based model. **a** Distribution of outcomes for different values of cost and turnover for an initial tumour composition of $(n_O, f_R) = (25\%, 0.1\%)$ ($n = 1000$ replicates per condition). This shows that while cures are rare, in a number of cases longer tumour control is achieved by continuous therapy. "N/A" denotes that the tumour had neither progressed nor been cured within the 10 year simulation time frame. **b** Outcome distributions for the same conditions as **a** but with a higher initial cell density ($(n_O, f_R) = (50\%, 0.1\%)$; $n = 1000$ replicates per condition). Inferior results for adaptive therapy become rarer, indicating a stochastic origin of these failures. **c** Treatment trajectories for the case with the worst outcome for adaptive therapy in **a** ($(n_O, f_R, c_R, d_T) = (25\%, 0.1\%, 30\%, 30\%)$; $n = 1$ replicate). **d** Simulation snapshots corresponding to **c**, showing that continuous therapy in this case progresses more slowly because all but one resistant nest goes extinct. **e** Treatment trajectories for the case with the worst outcome for adaptive therapy in **a**, but in the absence of turnover or cost ($(n_O, f_R, c_R, d_T) = (25\%, 0.1\%, 0\%, 0\%)$; $n = 1$ replicate). **f** Simulation snapshots corresponding to **e**, suggesting that in this case the branching pattern induced by competition during adaptive therapy is the reason for faster progression (see also Supplementary Fig. 9b for further analysis).

**distributions of resistance**. Having illustrated the theoretical implications of our work, we will focus in the final part of this paper on what these theoretical insights may teach us about intra-tumoral competition in patients. Androgen deprivation therapy is an integral part of prostate cancer treatment as many tumours are initially dependent on androgen signalling for their growth[50]. The inevitable development of androgen-independence, as well as the impact of treatment on quality of life, has motivated a number of intermittent therapy trials in prostate cancer (e.g., refs. [42,51,52]). The trialled algorithms administer treatment until the levels of PSA, a blood-based biomarker used for tracking tumour burden, are reduced to normal levels. Subsequently, treatment is withdrawn until PSA levels again exceed some upper limit, when treatment is reinstated. Interestingly, it is observed that some patients are cycling rapidly under treatment whereas others are cycling more slowly (Fig. 7a), although the relationship between cycling frequency and outcome is unclear[42].

Using the ODE model (Equations (2)–(4)) we previously showed that we could explain the different cycling dynamics across patients in the Phase II study by Bruchovsky et al.[42] by different values of the rate of tumour turnover and the resistance cost, suggestive of different underlying disease biology[32]. In order to test for possible differences in the tumours' spatial architectures, we fitted the ABM to these same data, which consist of monthly PSA measurements of 65 patients undergoing intermittent androgen deprivation treatment for recurrent, locally advanced prostate cancer. We allowed the initial tumour composition, as characterised by the proximity to carrying capacity ($n_0$) and the initial resistance fraction ($f_R$), the level of resistance costs ($c_R$), and the rate of cell turnover ($d_T$) to vary on a patient-specific basis, whilst keeping all other parameters fixed across the cohort at the values shown in Table 1. We also explored varying the proliferation rate ($r_S$) instead of turnover, but this yielded poorer model fits (not shown) and so we will here focus on the results when varying turnover. Given the stochastic nature of our simulations we assembled the fit for each patient from the mean of 25 independent stochastic replicates.

We find that, despite the simplicity of our model, it can recapitulate the cycling dynamics for a majority of patients, including both fast and slowly cycling patients (Fig. 7a; see also Supplementary Fig. 2). Moreover, fast and slow cyclers are associated with distinct spatial dynamics in our simulations (Fig. 7b and Supplementary Movie 3). In fast cycling patients, the model predicts a "carpet-like" structure with many small, independent, sensitive and resistant nests (Patient 75). In contrast, in slowly cycling patients growth is driven by only a handful of large, "patch-like", colonies (Patient 88). In order to understand which parameters are key in driving this behaviour, we fitted the model keeping either the initial conditions ($n_0$ and $f_R$) or the cell kinetic parameters ($c_R$ and $d_T$) fixed across the cohort. We find that allowing just cost and turnover to be patient-specific can explain the data almost as well as the full 4-parameter model (Supplementary Figs. 3 & 10). In contrast, the model assuming that inter-patient variability is caused by different initial conditions fits poorly (Supplementary Fig. 10).

A key result we obtained when we analysed these data using the ODE model was that the patients' cycling speed was correlated with the fitted values of resistance cost and turnover[32]. Fast cycling patients were associated with high levels of cost and low turnover, and conversely for slowly cycling patients, indicating different underlying disease biologies. That being said, the non-spatial nature of the ODE model made it difficult to interpret how exactly these differences may manifest themselves in practice. To investigate whether its spatially explicit nature could provide us with additional biological insight, we repeated this analysis with the ABM. In agreement with the ODE work, we find a negative correlation between cost and turnover, both when fitting all four parameters (Supplementary Fig. 11), and when fitting only cost and turnover (Pearson's correlation coefficient: $r_{56} = -0.76$, $p = 1.4 \times 10^{-11}$; Fig. 7c). Furthermore, also in the ABM fast cyclers are associated with large values of cost and small values of turnover, and vice versa for slow cyclers, supporting the hypothesis of different underlying biologies (Fig. 7c). Importantly, we can now further characterise these biologies (Fig. 7d and Supplementary Movie 4): In fast cyclers, where turnover is low and cost is high, most resistant cells present at the start of treatment will survive, but only expand very slowly, which yields a diffuse, carpet-like appearance of these tumours. In contrast, in slow cyclers, where turnover is high and cost low, many initially present resistant colonies will go extinct, but those that do survive will be able to expand more rapidly. This creates a more defined, patch-like appearance. Repeating our analysis with cells seeded as densely packed disks in the centre of the domain and allowed to expand outwards corroborates these conclusions (Supplementary Fig. 12).

Moreover, whilst there is qualitative agreement between the ODE model and the ABM, we find that additional insight can be gained from studying when, and why, they disagree. For most patients the goodness-of-fit for both models is comparable (in 45/65 patients the difference in $r^2$ is less than 0.1; Supplementary Fig. 13a), but the ODE model generally fits slightly better, in particular for the peak PSA values of fast cycling patients (Supplementary Fig. 13b, c). This can be explained by the fact that both models assume the same cell proliferation rate but differ in their predictions of the fraction of cells that is actually dividing, so that the population growth in the ABM is slower than that in the ODE. However, simply increasing the proliferation rate does not improve the ABM fits either as this results in a much higher rate of drug kill than is seen in the data, due to the way in which treatment feeds back on the tumour's spatial structure, and due to our assumption that the drug acts during cell division (not shown). Together, this suggests that the regrowth dynamics in the patients is more consistent with an exponential growth model than with a 2-D surface growth model, or that the treatment may also kill cells in a cycle-independent fashion.

A further interesting observation is that no patients are to be found in the top left corner of the graph, and only one patient is located in the bottom right corner (Fig. 7c). This can be explained by the trial's patient selection criteria. When cost and turnover are small, response is weak so that patients in this parameter regime would not have been able to produce the initial PSA normalisation required for study inclusion. Conversely, patients in whom both cost and turnover are high will show durable responses and are, thus, unlikely to be refractory after initial therapy (recall Fig. 3a for how cost and turnover impact response). Moreover, previously, we observed that patients who progressed on the trial were characterised not by a lack of cost or turnover but by a smaller combination of the two[32]. Our current analysis corroborates this context-dependence of the resistance costs. Progressors (yellow circles) cluster along the upper boundary of the line of fits (Fig. 7c). Accordingly, we detect no statistically significant difference in the turnover estimate (Mann–Whitney test, $U_{47,9} = 210$, $p = 0.49$), but a significant difference in the sum of the two ($c_R + d_T$; Mann–Whitney test, $U_{47,9} = 45$, $p = 1.1 \times 10^{-4}$). That being said, we do observe a statistically significant difference in the estimated cost values (Mann–Whitney test, $U_{47,9} = 95$, $p = 4.8 \times 10^{-3}$), which indicates that the resistance cost may play more of a role in the spatial ABM than it did in the non-spatial ODE model (for a more in-depth comparison of the ABM and ODE model parameter estimates, see Supplementary Fig. 13d).

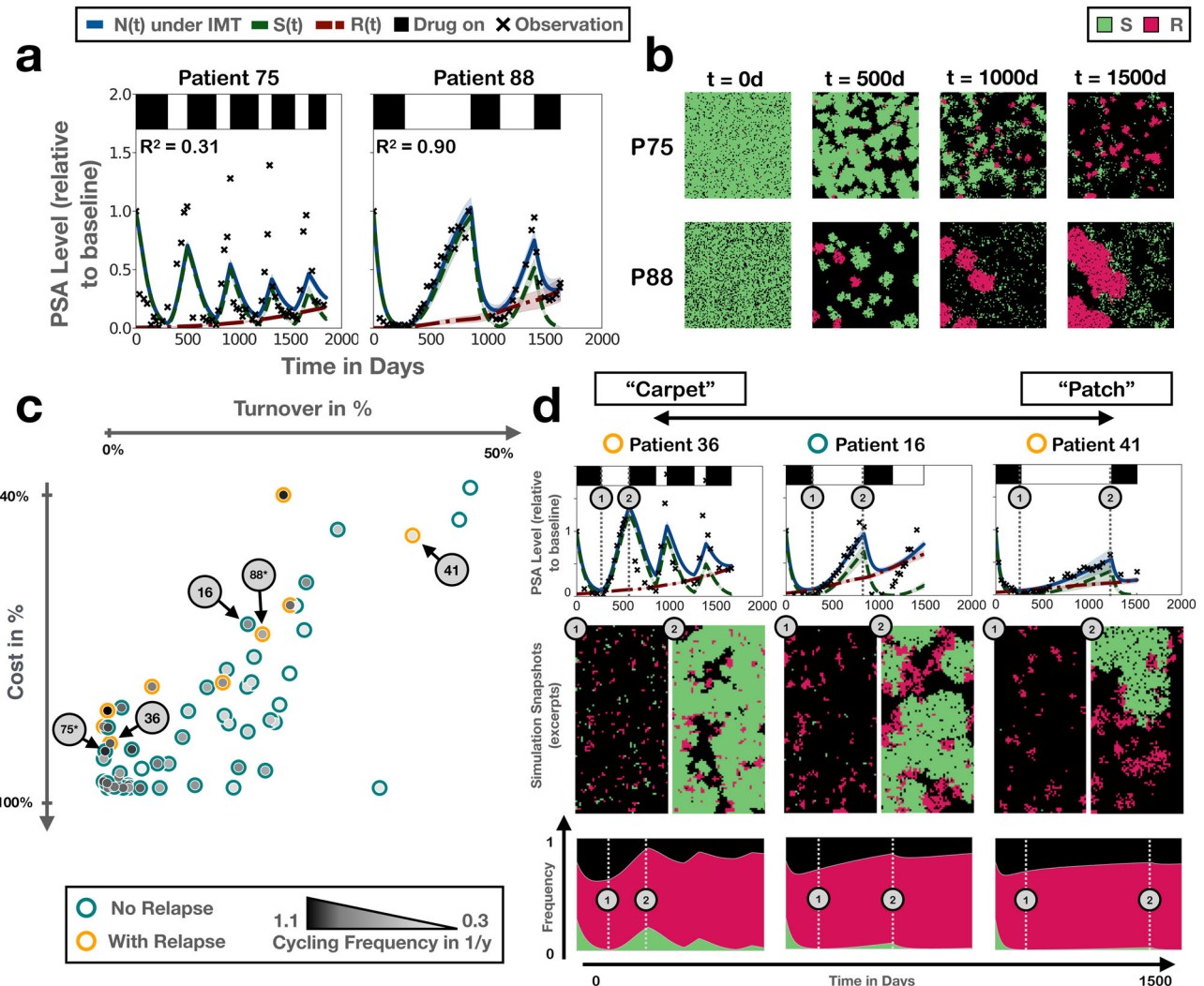

**Fig. 7 Analysis of the cycling dynamics of 65 prostate cancer patients undergoing intermittent androgen deprivation therapy in the trial by Bruchovsky et al.[42].** IMT: intermittent therapy, PSA: prostate-specific antigen, ABM: agent-based model. **a** Representative fits of the ABM to the PSA data from a fast and a slow cycling patient (Patients 75 and 88, respectively). Note that $N(t)$, $S(t)$, and $R(t)$ have been normalised relative to $n_0$. **b** Snapshots from one of the simulations in **a**, showing distinct patterns of resistance growth in the two patients, where in Patient 75 resistance emerges from many small colonies, whereas in Patient 88 it is driven by only a few, but rapidly expanding, nests. **c** Negative correlation between the estimated values of cost and turnover, revealed when fitting just cost and turnover on a patient-specific basis. Note that while Patients 75 and 88 are marked for reference, their fits here are not those shown in **a**, which were fitted with $n_0$, $f_R$, $c_R$, and $d_T$ being allowed to vary. For an overview of all fits for this 2-parameter model, see Supplementary Fig. 3. **d** Treatment trajectories, simulation snapshots and neighbourhood composition, illustrating the dynamics for patients in different areas of the parameter space in **c** ranging from what we term "carpet"-like appearance (many small, resistant colonies) to "patch"-like appearance (few, but large, resistant colonies; see also Supplementary Movie 4). Throughout the figure, lines and shading denote the mean and standard deviation of $n = 250$ independent replicates, respectively.

Finally, we sought to understand how the proposed differences between the inferred spatial architectures of fast and slow cycling patients may affect intra-tumoral competition. This shows that in slowly cycling patients essentially all competition is intra-specific, whereas in fast cyclers competition with sensitive cells plays more of a role (Fig. 7d). Overall, this supports our hypothesis that different cycling speeds reflect different underlying disease biology[32], and suggests this may not only be driven by differences in the cell kinetics but also may manifest itself in distinct tumour architectures and competition landscapes.

## Discussion

The aim of this study was to better understand competition for space during adaptive therapy. To do so, we studied a simple 2-D,

on-lattice ABM in which tumour cells were classified as either drug-sensitive or resistant. Leveraging the individual-based nature of our model we explicitly measured, for the first time, spatial competition between cells during therapy. This allowed us to visualise and quantify how treatment breaks during adaptive therapy increase the competitive inhibition of resistant cells. Furthermore, we capitalised on this to explore how different model parameters, which have previously been shown to modulate the benefit of adaptive therapy, impact competitive suppression. For example, we showed how reducing the initial cell density diminishes suppression, whereas a higher initial resistance fraction results in a similar level of inhibition per adaptive therapy cycle, but fewer cycles.

Moreover, this analysis revealed that intra-specific competition of resistant cells with each other is an important, but so far

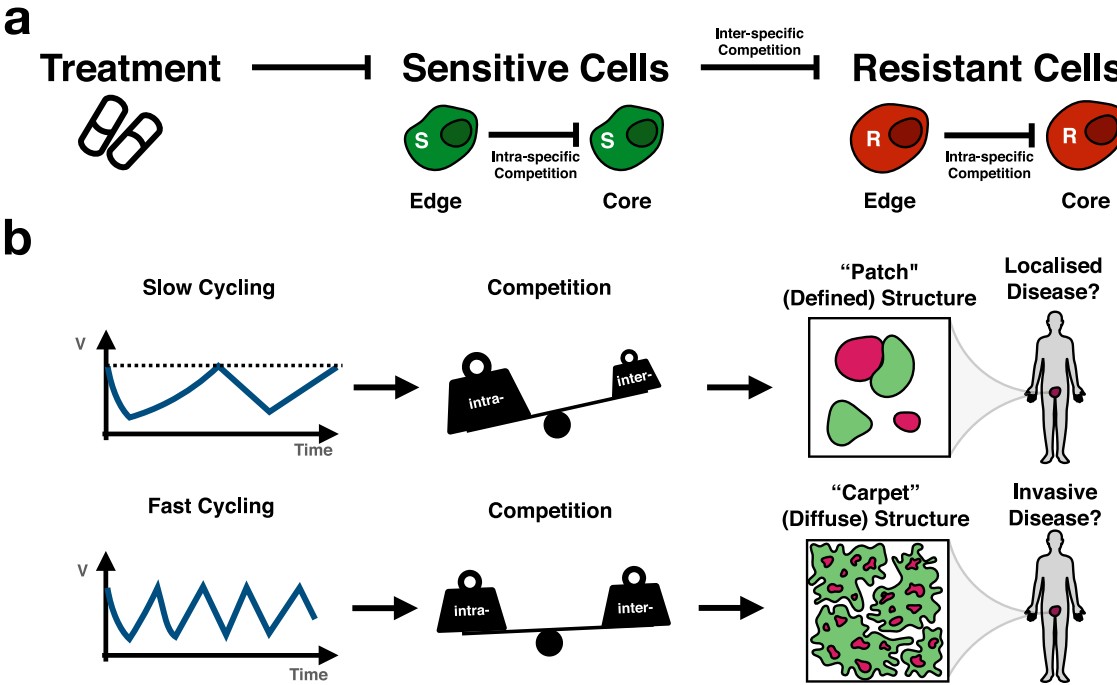

**Fig. 8 Paper summary: the tumour's spatial architecture determines the ratio of intra- to inter-specific competition, and thereby the response dynamics under adaptive therapy. a** The current theory around adaptive therapy postulates that it keeps resistant cells in check by inter-specific competition from sensitive cells, which can be controlled via treatment ([9,11]; adapted from ref. [32]). Here, we show that intra-specific competition within each population is a further important factor. **b** In addition, our analysis suggests that the more diffusely growing a tumour, the greater the fraction of inter-specific competition. In addition, this may be reflected in faster cycling frequency under intermittent treatment. Going forward, we propose that incorporating knowledge of the tumour's spatial architecture and resulting competition structure will help to design more effective adaptive therapy strategies.

overlooked, factor in adaptive therapy. As a resistant cell divides, most of its daughters will end up being trapped in the central core of the resistant colony, able to divide only upon the death of another resistant cell (or upon migration out of the core). As such, adaptive therapy is most effective when resistance is clustered in a single location, and surrounded by sensitive cells, because it can leverage inter-specific competition at the edge of the resistant colony to maximise intra-specific competition between resistant cells at the core (Fig. 8a).

An important implication of this observation is that it matters how resistance is distributed across the tumour. If resistance arises in a single location then it can be controlled more effectively with adaptive therapy than if resistance is present at multiple sites, either within the same lesion, or at different metastatic sites within the body. As such, we extend previous results by Bacevic et al.[14] and Gallaher et al.[23]. This theoretical finding also agrees with recent experimental evidence from Colom et al.[53] where it was shown that clones with oncogenic potential can keep each other in check through competition if they arise in close vicinity, but will expand into neoplasms if located further apart (or in isolation). Moreover, we show that the intra-tumoral competition, and thus treatment dynamics, changes with the tumour's spatial organisation. As such, care will be required when using non-spatial ODE models to make quantitative predictions about the benefit of adaptive therapy. Specifically, when we compare the results from the ABM with its corresponding non-spatial ODE model, we observe that while adaptive therapy is superior in both models, its relative benefit compared to continuous therapy is smaller in the ABM, and there is less gain from the presence of resistance costs and turnover. This is because the two models make different assumptions about spatial competition. In the ODE model, cells are assumed to be perfectly mixed,

so that all cells experience the same competitive growth inhibition, which is simply a linear function of the total cell density. In contrast, the lack of migration in the ABM results in spatial segregation of different colonies, so that the competitive inhibition experienced by a cell depends on the cell's local neighbourhood, and varies across the tumour. Consequently, sensitive cells in the ODE model will always be able to competitively suppress resistant cells, whereas in the ABM this is only possible if the sensitive colony grows in close vicinity to the resistant colony. This indicates that a detailed understanding of intra-tumoral competition is required, in order to determine whether or not a patient will receive a clinically meaningful gain from adaptive therapy. This point is supported by recent work by Viossat and Noble[19], and Farrokhian et al.[54], that found that while different ODE models of adaptive therapy agree qualitatively, there are significant differences in their quantitative predictions depending on how competition is modelled.

But how may we infer the spatial distribution of resistance? Tissue biopsies would provide the most direct and detailed measurements, but are invasive and often impractical. As an alternative, we propose that it may be possible to use mathematical modelling to gather spatial insights from the patient's longitudinal response dynamics (Fig. 8b). When fitting our ABM to PSA data from prostate cancer patients undergoing intermittent androgen deprivation therapy we find that the speed at which patients cycle between treatment on- and off-periods correlates with distinct forms of spatial organisation of the tumours in our simulations. Fast cyclers are associated with more diffuse ("carpet"-like) tumours whereas slow cyclers are associated with more compact ("patch"-like) tumours (Fig. 8b). In an analysis of their trial data, Bruchovsky et al.[42] reported a "suggestive trend that a Gleason score < 6 may be associated with a

slightly longer time off treatment in the initial 2 cycles". Interestingly, lower Gleason scores indicate a more defined, glandular tissue architecture, which would be consistent with our model predictions. That being said, since the ABM model does not fully capture the PSA peaks in fast cycling patients, requires small turnover values of less than 1% in more than 10% of patients, and cannot fit some patients at all, this result should be taken with a grain of salt. In particular, these observations suggest that the growth dynamics in the patients appears to be faster than that in the ABM. This may be because these tumours grow in 3-D, which provides more space for interactions than the 2-D lattice in the ABM. Nevertheless, our work shows how using a spatial model may allow us to gain understanding of the tissue architecture and competition landscape within a tumour - even if this comes at the cost of slightly poorer fits in comparison to an ODE model. Going forward, we plan to investigate this idea further with a more prostate cancer-specific, 3-D model.

In addition, it may also be possible to gather some information about the spatial distribution of resistance from the characteristics of the resistant population. In particular, if resistance is driven by a single clone, then it will likely be initially confined to a single, or at most a small number of, sites within the tumour. In contrast, if resistance is driven by multiple clones, as has, for example, been observed in colorectal cancer[55], then it is likely to exist in multiple locations simultaneously. Liquid biopsies are showing promise at detecting and characterising the clonality of emerging drug resistance and, as such, may provide a useful tool for informing adaptive therapies[33,55].

While we cannot easily alter it, understanding the spatial distribution of resistance in a patient may be relevant in the design of adaptive treatment schedules. Gallaher et al.[23] found in an off-lattice ABM that the rate of cell migration (and therefore of spatial mixing) determined whether a modulation-based adaptive algorithm (treatment is modulated in small increments, rather than withdrawn), or a vacation-based algorithm (treatment is either on or off) was more effective. In particular, spatially confined tumours favoured modulation, whereas in invasive tumours more benefit was derived from the vacation-based strategy[23]. Conversely, Benzekry and Hahnfeldt[56] concluded from the study of a combined ODE and partial differential equation model that metronomic chemotherapy scheduling (low dose, high frequency) may be more effective in controlling metastatic disease than aggressive standard-of-care treatment (high dose, low frequency). Investigating how to best adapt treatment when there are multiple resistant nests, and/or metastasis, is an important direction of future research.

A final observation we make is that there can be significant variation in the benefit of adaptive therapy between stochastic replicates of our simulations, despite identical model parameters. In fact, in some cases longer tumour control is achieved by continuous, and not adaptive, therapy. Variance depends on the number of resistant cells initially present in the simulations, and their distribution, which further highlights the importance of the spatial distribution of drug resistance within the tumour. Moreover, even though resistance costs and turnover increase the average benefit of adaptive therapy, they also increase variability in outcomes. As a result, the greater the benefit of adaptive therapy on average the more we may see variation between individual patients, with some gaining much more time, and some less, than expected. While our simulations are not suited to make quantitative statements about the magnitude of this problem due to the unrealistically small cell numbers, Hansen et al.[57] have recently raised similar concerns. Thus, we advocate further study of the impact of inter-patient variability in adaptive therapy using, for example, Phase i trials[58], in order to inform future clinical trial design. Moreover, the idea that competition during

adaptive therapy may alter the tumour morphology warrants further investigation, as Enriquez-Navas et al.[13] have found that adaptive therapy changes tumour vascularity in vivo, and several studies have linked tumour morphology to outcome (see e.g., refs. [59,60]).

In aiming to keep our model tractable we have made a number of simplifying assumptions. We assumed no movement and no pushing of cells, which has been shown by Gallaher et al.[23] and Bacevic et al.[14] to reduce the benefit of adaptive therapy, as it allows resistant cells to squeeze through surrounding sensitive cells. Moreover, for computational reasons, we restricted our analysis to a 2-D setting, which is arguably more representative of in vitro cell culture than a 3-D human tumour. We hypothesise that the extra dimension will hinder tumour control as it will allow resistant cells to more easily find space into which they can divide. That being said, we have also neglected the role of non-tumour tissue, which acts as an additional competitor for space and resources in the tumour, and may help to control resistant subpopulations[21]. The recent paper by M A et al.[61] takes a first step in this direction, independently corroborating not only some of our results but also exploring the impact of fibroblasts. We, and others, have also investigated the important role of metabolism in regulating tumour progression, immune dynamics and treatment response[62–64]. The role of tumour intrinsic metabolism versus extrinsic tissue microenvironment metabolism has also not been considered here. However, their inclusion is more likely to enhance the impact of adaptive therapy than diminish it due to the potential for increasing the cost of resistance (e.g., refs. [32,65–67]).

A final, important caveat is that we have not explicitly modelled the mechanism by which resistance arises. Depending on whether it arises through mutation, phenotypic switching, or is environmentally induced, this may drive different initial distributions of resistant cells, and will also result in different dynamics during treatment due to de novo resistance acquisition (see also refs. [19,22,23,48]). Furthermore, our model cannot explain how some of the initial tumour compositions we have analysed would have arisen prior to treatment. For example, if resistance costs and turnover are assumed to be high, then resistance will disappear in our simulations if the tumour is left untreated (not shown).

To conclude, we want to highlight parallels with work carried out on antibiotic resistance, which suggest possible areas of synergy and avenues for future research. Also in bacteria it has been demonstrated that sensitive cells can slow the growth of resistant cells, even in the spatially relatively unconstrained environment of a bioreactor[8,68]. Similarly, Fusco et al.[30] found in 2-D in vitro biofilms that they could delay the release of resistant bacterial mutants via a reduction in the drug dose, which kept them trapped in the centre of the colony for longer. As such, our results may be relevant also for the treatment dynamics in bacterial biofilms and, in fact, the 2-D nature of these systems may make these useful experimental model systems to test some of our theoretical findings. At the same time, this research has revealed that the packed environment of a biofilm may not necessarily hinder, but can in fact aid, resistance evolution. "Gene surfing" at the edge of the expanding colony can allow even late occuring (potential resistance) mutations to play an important role in the colony's evolutionary trajectory[30], and mechanical interactions in growing films can alleviate resistance costs because slower growing cells can be pushed along by their neighbours[31]. Moreover, where resistance is conveyed by enzymatic digestion of the drug, cooperative interactions between sensitive and resistant cells may evolve in which the precise spatial arrangement of the cells in the biofilm plays an important role[69,70]. Such facilitation has recently also been reported in cancer[71,72] and strongly

motivates extending our research to include cooperative interactions, and to ask how they may be tackled using multi-drug strategies[24,25,73–75].

To sum up, in this paper, we have consolidated and advanced our understanding of how competition between tumour cells may be leveraged by careful treatment modulation. We have shown that the spatial organisation of resistant populations is an important, and understudied, factor in cancer treatment. This strengthens the argument for patient-specific, adaptive therapy protocols that explicitly consider not only a tumour's evolution but also its ecology.

## Data availability

No new experimental or clinical data were generated in this study. All source data for the main figures within this study are available on Figshare[76], and the code to generate each figure has been deposited on Zenodo at ref. [41]. The patient data were downloaded from the author's webpage at http://www.nicholasbruchovsky.com/clinicalResearch.html in July 2020.

## Code availability

The code for the computational model and for all analyses presented in this paper is publicly available at https://github.com/MathOnco/strobl2021_space_modulates_competition_AT, and has been archived on Zenodo at ref. [41].

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

## Acknowledgements

We thank Prof. Ruth Baker, Dr. Gregory Kimmel, and Dr. Etienne Baratchart for helpful discussions about the implementation of the stochastic simulation algorithm. We also thank Dr. Bruchovsky and his team for making the data of their Phase II study publicly available, and Prof. Kazuyuki Aihara, Jonas Abersbach, Prof. Laurence Klotz, and Prof Larry Goldenberg for addressing our questions and maintaining this unique resource after the passing of Dr. Bruchovsky. Finally, we thank three anonymous reviewers for their detailed feedback, which helped to improve this paper. M.S. was supported by funding from the Engineering and Physical Sciences Research Council (EPSRC) and the Medical Research Council (MRC) [grant number EP/L016044/1]. A.A. and M.R.T. gratefully acknowledge funding from both the Cancer Systems Biology Consortium and the Physical Sciences Oncology Network at the National Cancer Institute, through grants U01CA232382 and U54CA193489 as well as support from the Moffitt Center of Excellence for Evolutionary Therapy.

## Author contributions

Conceptualisation, M.S., P.M. and A.A.; methodology, M.S., J.G. and J.W.; software, M.S.; formal analysis, M.S., J.G., J.W. and M.R.T.; investigation, M.S., J.G., J.W., M.R.T.; resources, P.M. and A.A; data curation, M.S.; writing—original draft preparation, M.S.; writing—review and editing, J.G., J.W., M.R.T., P.M. and A.A.; visualisation, M.S., J.G. and J.W.; supervision, P.M. and A.A.; project administration, P.M. and A.A.; funding acquisition, M.R.T., P.M. and A.A. All authors have read and agreed to the published version of the manuscript.

## Competing interests

The authors declare no competing interests. The funders had no role in the design of the study; in the collection, analyses, or interpretation of data; in the writing of the manuscript, or in the decision to publish the results.
