## [Peer Review File · Communications Medicine]

This manuscript has been previously reviewed at another Nature Portfolio journal. This document only contains reviewer comments and rebuttal letters for versions considered at Communications Medicine.

Reviewers' comments:

Reviewer #1 (Remarks to the Author):

I'm re-reviewing this paper after revisions (Reviewer #1). In general, I still view this paper in a favorable light and think it should be published, especially after they've augmented the MS with some nice additional analyses.

At this point, my major remaining reservation is that I would like the authors to be more upfront about some of the model's shortcomings. It would be useful to create discuss in the main text the issues highlighted in 1c, 1e, 1h.

1a. Thank you for adding clarifications about the exclusion criteria and R^2 fits to the text. No further comments.

1b. Thanks for the clarification here. No further comments.

1c. In my opinion, this is not sufficient explanation about this point: "our inferred parameter values should be viewed with some reservations." All inferred model parameters should be viewed with some reservations. Below (1d), the authors argue that the death rate is the correct biological parameter to fit (as opposed to the birth rate) and it's capturing something other than simple cycling. I would like to see the authors state much more explicitly how the reader is supposed to interpret their fits that $d \approx 0$ in so many instances. If we don't trust the quantitative parameter fits, why should we trust the spatial signatures they create?

1d. Thanks for performing this extra analysis. I agree that the birthrate analyses are interesting, and while the fits do not appear as good as the death rate varying model for the plotted example, I stand by my initial point. It still feels to me that the model has one lever to rescale time, and so I am unsurprised that fast and slow cycling individuals have different inferred death rates, and that different death rates create different spatial configurations. I'm still not convinced that huge differences in death rates explain the patient data, but I accept that they're what makes the most sense under the presented model.

1e. I appreciate this additional interesting and important analysis, and the authors' discussion of it in the supplemental figure legend. I feel that these are important points, and they are not adequately represented in the main text: "(for a more in-depth comparison of the ABM and ODE fits, see Figure A13)." I understand not wanting to derail the MS, but given how connected this model is to the authors' previously published one and how much time the authors spend discussing how the two relate, I do think that the models should be directly compared somewhere where the majority of people who are reading the MS will encounter it. I think it's worth saying simply that even though the ABM fits are slightly worse than the ODE model, understanding how the competition dynamics might be operating in space is still useful - maybe even upfront in lines 574-576.

1g. Thanks for this analysis. No further comments.

1h. Thanks for this explanation. I think the PSA spike fits in fast cycling patients and their potential implications about the drug kill model should be discussed in the text as a model shortcoming.

2. I'm satisfied on this point.

3. Thanks for this additional analysis. No further comments, except that it feels like there is potentially a lot more to analyze here, but I agree that it's beyond the scope of the MS.

4. Thanks for this additional analysis - it does indeed help explain the behaviors, and I think reveals a fruitful new direction for future work to understand a little bit more concretely what's happening.

Reviewer #2 (Remarks to the Author):

My concerns have been well addressed by the authors. Although the model still seems to be over-simplified, it might stimulate discussions and further studies on developing a better spatial model of AT. I'm happy to see this manuscript to be available.

Thanks for your email and asking me about my comments on the authors' response to Reviewer 3's concerns. After carefully reading Reviewer 3's concerns, I agree with him/her raising that "there do not seem to be many important new results in terms of the analysis of the model that makes up the bulk of the study". However, it's known that modeling spatial structure in the context of cancer treatment and fitting it to real data is notoriously difficult. The current manuscript by Strobl and colleagues provided a simple spatial modeling framework to this end. Although most the insights generated by the their model and data fittings are not new, I think it's still helpful for the field of AT and might stimulate further studies to develop a better model of AT. Also I think the authors have addressed Reviewer 3's concerns well.

Reviewers' comments:

Reviewer #1 (Remarks to the Author):

I'm re-reviewing this paper after revisions (Reviewer #1). In general, I still view this paper in a favorable light and think it should be published, especially after they've augmented the MS with some nice additional analyses.

At this point, my major remaining reservation is that I would like the authors to be more upfront about some of the model's shortcomings. It would be useful to create discuss in the main text the issues highlighted in 1c, 1e, 1h.

1a. Thank you for adding clarifications about the exclusion criteria and R^2 fits to the text. No further comments.

1b. Thanks for the clarification here. No further comments.

1c. In my opinion, this is not sufficient explanation about this point: "our inferred parameter values should be viewed with some reservations." All inferred model parameters should be viewed with some reservations. Below (1d), the authors argue that the death rate is the correct biological parameter to fit (as opposed to the birth rate) and it's capturing something other than simple cycling. I would like to see the authors state much more explicitly how the reader is supposed to interpret their fits that $d \approx 0$ in so many instances. If we don't trust the quantitative parameter fits, why should we trust the spatial signatures they create?

This is a good point, and we have revised our discussion of these results in the discussion section to address these points. We think the reason for the $d \approx 0$ is that the growth dynamics in the patients is faster than that which can be produced by the ABM due to the strong spatial restrictions of the 2-d lattice. This is why the ODE also fits slightly better than the ABM, in particular for the fast cycling patients. Nevertheless, we think that there is support for the spatial signatures we propose as i) the death rate-driven model fits the data better than the birth rate-driven model, and ii) these predictions are consistent with an observation made by Bruchovsky et al, that a lower Gleason score correlated with slower cycling.

1d. Thanks for performing this extra analysis. I agree that the birthrate analyses are interesting, and while the fits do not appear as good as the death rate varying model for the plotted example, I stand by my initial point. It still feels to me that the model has one lever to rescale time, and so I am unsurprised that fast and slow cycling individuals have different inferred death rates, and that different death rates create different spatial configurations. I'm still not convinced that huge differences in death rates explain the patient data, but I accept that they're what makes the most sense under the presented model.

1e. I appreciate this additional interesting and important analysis, and the authors' discussion of it in the supplemental figure legend. I feel that these are important points, and they are not

adequately represented in the main text: "(for a more in-depth comparison of the ABM and ODE fits, see Figure A13)." I understand not wanting to derail the MS, but given how connected this model is to the authors' previously published one and how much time the authors spend discussing how the two relate, I do think that the models should be directly compared somewhere where the majority of people who are reading the MS will encounter it. I think it's worth saying simply that even though the ABM fits are slightly worse than the ODE model, understanding how the competition dynamics might be operating in space is still useful - maybe even upfront in lines 574-576.

Thank you for this suggestion. We have integrated a discussion of these results in Section 3.10 as suggested, and into the discussion.

1g. Thanks for this analysis. No further comments.

1h. Thanks for this explanation. I think the PSA spike fits in fast cycling patients and their potential implications about the drug kill model should be discussed in the text as a model shortcoming.

We have added a discussion of this observation to Section 3.10. in the results section, and to the discussion.

2. I'm satisfied on this point.

3. Thanks for this additional analysis. No further comments, except that it feels like there is potentially a lot more to analyze here, but I agree that it's beyond the scope of the MS.

4. Thanks for this additional analysis - it does indeed help explain the behaviors, and I think reveals a fruitful new direction for future work to understand a little bit more concretely what's happening.

Reviewer #2 (Remarks to the Author):

My concerns have been well addressed by the authors. Although the model still seems to be over-simplified, it might stimulate discussions and further studies on developing a better spatial model of AT. I'm happy to see this manuscript to be available.

Thanks for your email and asking me about my comments on the authors' response to Reviewer 3's concerns. After carefully reading Reviewer 3's concerns, I agree with him/her raising that "there do not seem to be many important new results in terms of the analysis of the model that makes up the bulk of the study". However, it's known that modeling spatial structure in the context of cancer treatment and fitting it to real data is notoriously difficult. The current manuscript by Strobl and colleagues provided a simple spatial modeling framework to this end. Although most the insights generated by the their model and data fittings are not new, I think it's still helpful for the field of AT and might stimulate further studies to develop a better model of AT. Also I think the authors have addressed Reviewer 3's concerns well.

We thank Reviewer 2 for their thoughtful and balanced evaluation of our manuscript, and their encouraging words regarding the model fitting. We agree that the model is very simple, but we would argue that this simplicity allows us to obtain an in-depth understanding of which assumptions are the most impactful in influencing the dynamics. We hope that this will help guide the study of more complex models in the future by helping to identify which extra complexities to consider in these studies.

REVIEWERS' COMMENTS:

Reviewer #1 (Remarks to the Author):

Thank you for these edits. All of my concerns have been addressed.